# Interpretable Generalized Additive Models for Datasets with Missing Values

**Hayden McTavish**∗
Department of Computer Science
Duke University
Durham, NC 27705
hayden.mctavish@duke.edu

**Jon Donnelly***
Department of Computer Science
Duke University
Durham, NC 27705
jon.donnelly@duke.edu

**Margo Seltzer**
Department of Computer Science
University of British Columbia
Vancouver, BC V6T 1Z4
mseltzer@cs.ubc.ca

**Cynthia Rudin**
Department of Computer Science
Duke University
Durham, NC 27705
cynthia@cs.duke.edu

## Abstract

Many important datasets contain samples that are missing one or more feature values. Maintaining the interpretability of machine learning models in the presence of such missing data is challenging. Singly or multiply imputing missing values complicates the model's mapping from features to labels. On the other hand, reasoning on indicator variables that represent missingness introduces a potentially large number of additional terms, sacrificing sparsity. We solve these problems with M-GAM, a sparse, generalized, additive modeling approach that incorporates missingness indicators and their interaction terms while maintaining sparsity through $\ell_0$ regularization. We show that M-GAM provides similar or superior accuracy to prior methods while significantly improving sparsity relative to either imputation or naïve inclusion of indicator variables.

## 1 Introduction

Interpretability is essential for a wide range of machine learning applications Rudin et al. (2022). Missing data pose a challenge to interpretability, because many simple models (e.g., linear models) are not well-defined when data are missing. This raises the question: how can interpretability be maintained for datasets with missing values?

We introduce an interpretable model class, M-GAM, that extends Generalized Additive Models (GAMs) to handle missing data. GAMs take the form of a linear combination of univariate component functions, with one function corresponding to each feature; this univariate nature is the core reason for their interpretability (Rudin et al., 2022). We introduce two sets of boolean variables for each feature. The first consists of missingness indicators that identify which features have missing values. The second consists of missingness adjustment terms that adjust the shape curves for other features for each missing features in a sample. This *maintains our ability to view a GAM as a sum of univariate shape functions even when modeling interactions with missing data*. As such, an M-GAM is much simpler to interpret than a GAM built on imputed data, since it avoids creating multivariate features as happens when imputing features from multiple others. This is illustrated in Figure 1.

---

∗These authors contributed equally to this work.

38th Conference on Neural Information Processing Systems (NeurIPS 2024).

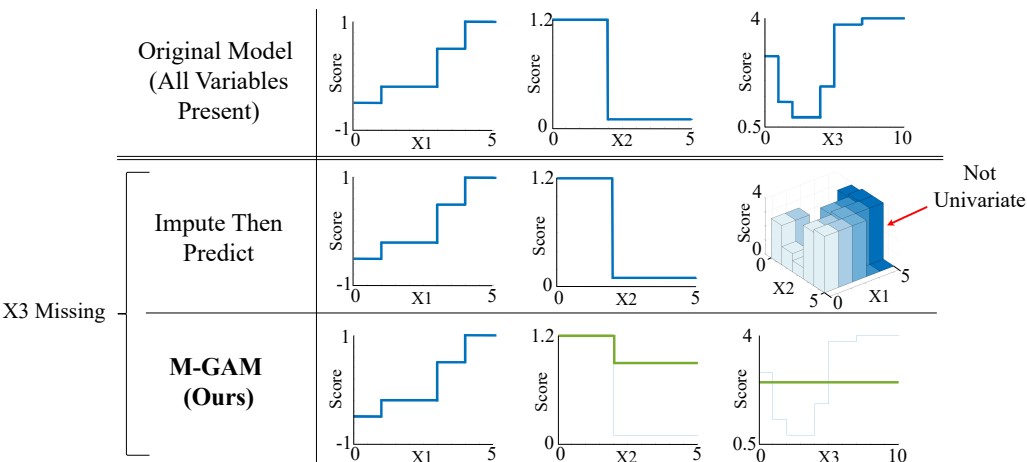

Figure 1: A comparison of how GAMs that use underlying imputation (middle row) and M-GAMs (bottom row) behave when a feature is missing. **Top**: When no data are missing, the overall output logit for both models is the sum of three univariate shape functions. **Middle**: When $X3$ is missing, it is imputed as $X3 = X1 + 2X2$, producing a 3D shape function that is difficult to understand. **Bottom**: M-GAM uses simple adjustments to existing univariate shape curves when $X3$ is missing (using the green curves instead of the light blue ones), making its reasoning process simple to follow. If the data were more than 3 dimensional, we would not be able to visualize the model with imputation, but M-GAM would still be easily visualized.

M-GAMs explicitly encourage sparsity. This reduces overfitting, which has been identified as a concern in prior work using missingness indicators (Van Ness et al., 2023), since realistic data may produce an overwhelming number of missingness indicators. Unlike prior methods that leverage missingness indicators, we use $\ell_0$ regularization (Liu et al., 2022; Dedieu et al., 2021; Hazimeh & Mazumder, 2020; Hazimeh et al., 2023), which directly optimizes for sparser, more interpretable models. Our ability to create sparse models allows us to include not only simple missingness indicators but also combinations of missingness indicator variables with M-GAM's missingness adjustment terms, without overfitting. Figure 2 illustrates an M-GAM fit on real data.

With this modeling framework, we make the following contributions: (1) We introduce M-GAM, a form of sparse generalized additive model that incorporates missingness directly into its reasoning. (2) We show that M-GAM provides substantial performance benefits relative to impute-then-predict models when synthetic missing-at-random (MAR) missingness is added to real datasets, while maintaining performance in real world settings with only naturally occurring missingness. (3) We show that M-GAM substantially reduces runtime relative to impute-then-predict methods built on multiple imputation while producing sparse, interpretable models.

## 2   Related Work

Missing data is a well studied problem in statistics. Traditionally, mechanisms by which data can be missing are organized into three categories: missing completely at random (MCAR), where missingness is independent of the value of all covariates; missing at random (MAR), where missingness in a variable $X_1$ is conditionally independent of the value of $X_1$ given all other variables; and missing not at random (MNAR), where missingness may depend on any variable (Little & Rubin, 2019).

For supervised learning, there are two common approaches to dealing with missing data: impute-then-predict, in which a standard machine learning model is fit on top of imputed data and used for prediction, and incorporating missing data handling directly in the predictive model.

A broad body of work studies imputation, particularly in the MAR setting – for a more thorough review, see Shadbahr et al. (2023). Imputation methods can broadly be sorted into single imputation methods (see Van Buuren, 2018, for a review of such methods), where each missing value is imputed once, and multiple imputation (Rubin, 1988; Van Buuren & Oudshoorn, 1999; Schafer &

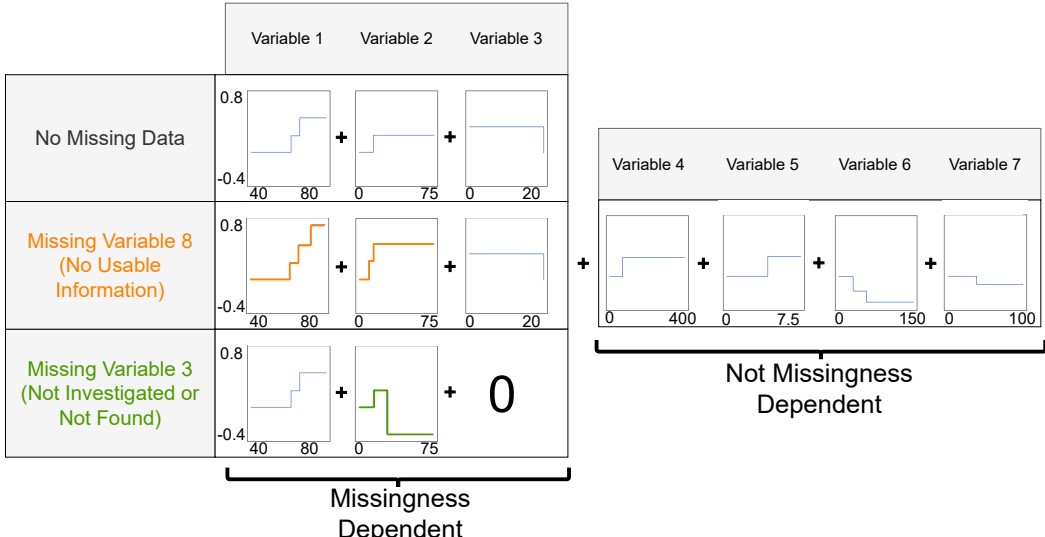

Figure 2: A generalized additive model (GAM) for the Explainable ML Challenge data from FICO et al. (2018) with missingness incorporated. This model handles missingness interpretably by explicitly providing alternative shape functions when a variable is missing. For example, in this model the shape function for variable 2 is adjusted when variable 3 is missing, and the shape function for variable 3 is removed. This model achieves comparable performance to convoluted black box approaches (such as random forests and/or MICE), but provides global interpretability (the entire model can easily be inspected) and local interpretability (the shape functions applied for a given sample can be easily visualized). An expanded version of this figure with variable names can be found in Appendix Figure 16. Shape functions in the right section are shared across all missing variable combinations. The type of missingness is indicated in parentheses next to the missing variable. Section F visualizes additional M-GAMs.

Graham, 2002; Stekhoven & Bühlmann, 2012; Mattei & Frellsen, 2019), where many alternative imputations are provided for each missing value. Multiple imputation is convenient because it integrates uncertainty into its imputations by providing a range of alternatives (Van Buuren, 2018).

Recent approaches directly incorporate missing data in the predictive model. Le Morvan et al. (2020) showed that, even when the target of prediction is a linear function, in the presence of missing data, the optimal model need not be linear in the original features. Rather, their optimal model was linear in the observed data *and* interactions between indicators for missing data and the observed data. Van Ness et al. (2023) showed that when missingness contains information about an outcome, linear models that directly include missingness indicators outperform models excluding this information. This inclusion of missingness indicators is especially recommended in the practical setting of predictive modeling, rather than the setting of statistical inference where MCAR, MAR and MNAR concepts are more commonly used (Sperrin et al., 2020).

There are also a wide variety of ad-hoc methods for handling missing data in tree-based models (Kapelner & Bleich, 2015; Twala et al., 2008; Beaulac & Rosenthal, 2020; Therneau et al., 1997) and boosting models (Wang & Feng, 2010; Chen & Guestrin, 2016) that involve prediction without explicit imputation. Beaulac & Rosenthal (2020) learn decision trees which avoid splitting on missing data when missingness follows a deterministic structure based on other known features. This sidesteps any need to query features when data is missing, but does not generalize to settings with less structured missingness. More generally, tree-based models can learn a branch direction for each split to use when the queried feature is unknown. This effectively imputes the response to the query, but keeps the model itself simple. This is the method used in XGBoost (Chen & Guestrin, 2016) and SKLearn's decision tree classifier (Pedregosa et al., 2011). Twala et al. (2008) and Kapelner & Bleich (2015) additionally incorporate the option to split on missingness itself, effectively encoding missingness as a value. Ding & Simonoff (2010) provides empirical support for incorporating this option to treat missingness as a value. We compare to XGBoost and SKLearn in our experimental section, finding that M-GAM better balances interpretability and performance, even when we allow missingness

indicator splits like in Twala et al. (2008); Kapelner & Bleich (2015); Ding & Simonoff (2010); Wang & Feng (2010). Most similarly to our own approach, Therneau et al. (1997) and Breiman (2017) discuss an alternative approach of surrogate splits: when a feature that is split on is missing, a set of other splits is used in place of the missing feature to evaluate the split. This practice of adjusting which features are used when one feature is missing bears some similarities to our use of missingness interaction splits that adjust the shape functions for some features when other features are missing, though these surrogate splits do not optimize for sparsity like M-GAM, and underperform more standard multiple imputation approaches Valdiviezo & Van Aelst (2015); Feelders (1999). Further work explores the idea of developing distinct models for use under different cases of missing features (Fletcher Mercaldo & Blume, 2020; Stempfle et al., 2023) or developing additive logical models with disjunctions, such that reliance on imputed values is low Stempfle & Johansson (2024).

It is critical to note that, for a dataset with $d$ features, adding indicators for missing data results in $2d$ features, and adding first order interactions between features and missingness results in $d(d-1) + 2d$ features. As such, without careful regularization, these models that explicitly handle missingness are complex and uninterpretable. This poses a challenge for their application in high stakes domains such as justice and medicine, where there have been calls to enshrine interpretability as a requirement for the use of machine learning methods (US Food and Drug Administration, 2021; European Commision, 2021). In contrast, M-GAM provides sparse, transparent models that handle missingness indicators and interactions by extending sparse generalized additive models (Liu et al., 2022). M-GAM provides an expressive model class for handling missingness while controlling the exploding number of missingness interaction terms through $\ell_0$ regularization.

## 3 Methodology

We denote a dataset of $n$ samples by $\mathbf{D} = (\mathbf{X}, \mathbf{y}) = \{(\mathbf{x}_i, y_i)\}_{i=1}^n$, where $\mathbf{x}_i \in (\mathbb{R} \cup \{\text{NA}\})^d$ is a $d$-dimensional vector of features, NA denotes a missing entry, and $y_i \in \{0, 1\}$ is our target label. We use $x_{i,j}$ to denote the $j$-th feature of the $i$-th sample. We use bold capital letters ($\mathbf{X}$) to denote matrices, bold lowercase letters to denote vectors ($\mathbf{x}_i$), capital letters to denote random variables ($X$), and lowercase letters to represent scalars ($x_{i,j}$). $\varepsilon$ denotes noise; any other Greek characters denote model parameters. We encode all binary comparisons to the value NA as $0$. That is, we follow the convention that $\mathbf{1}_{[\text{NA} \leq a]} = 0$ for any value $a \in \mathbb{R}$, where $\mathbf{1}_{[\cdot]}$ denotes the indicator function.

Note that, in practice, data are often missing for distinct yet identifiable reasons; for example, a measurement for one sample may be missing because it was never taken, while another may be missing because a researcher spilled coffee on the notes containing the data. As such, we explicitly consider distinct reasons for missing data. For a dataset with $c \in \mathbb{N}$ potential reasons for data to be missing, define the mapping $mcat : \mathbb{R} \cup \{\text{NA}\} \to \{0, 1, \ldots, c\}$ to map from an entry of $\mathbf{X}$ to a natural number indicating the reason that entry is missing ($0$ if the entry is not missing).

With notation established, we begin with a motivational proposition. Proposition 3.1 states that even if we can perfectly impute missing values, we may find greater predictive power by using missingness itself as a feature rather than by imputing missing values.

**Proposition 3.1.** *Let $I : (\mathbb{R} \cup NA)^d \to \mathbb{R}^d$ be an oracle imputation function that replaces all missing values in a vector with the correct non-missing entry. For a random variable $X \in \mathbb{R}^d$, let $f_1(X) := \mathbf{1}_{[\mathbb{E}[Y|I(X)]>0.5]}$ be the Bayes' optimal model using perfectly imputed data and $f_2(X) := \mathbf{1}_{[\mathbb{E}[Y|X]>0.5]}$ be the Bayes optimal model using missingness as a value. There exist data generating processes for $X$ and $Y$ where $P(Y = f_1(X)) < P(Y = f_2(X))$.*

Section A of the appendix provides a proof by construction for Proposition 3.1. The key insight behind Proposition 3.1 is that, when missingness is dependent on the label $Y$, missingness itself can be a powerful predictor of the label (this setting is called informative missingness in Van Ness et al., 2023). In particular, we can gain information about our label that is not available in other covariates (e.g., information from $\varepsilon_1$).

Proposition 3.1 may appear to conflict with Theorem 3.1 of Le Morvan et al. (2021), which states that a Bayes optimal model may be produced using impute-then-predict with almost any imputation model. This theorem hinges on the idea that, for most imputations, it is still possible to distinguish imputed data entries from non-missing entries. This is not the case for perfect imputation, which yields Corollary 3.2.

**Corollary 3.2.** *Let $\mathcal{R}(f, \mathbf{X}, \mathbf{y})$ denote the risk of a model $f$ for data $\mathbf{X}, \mathbf{y}$, and $\mathcal{R}^*$ the optimal risk. Let $I : (\mathbb{R} \cup NA)^d \to \mathbb{R}^d$ denote the oracle imputation function of Proposition 3.1. Under perfect imputation, it is possible for there to be no Bayes optimal model built on imputed data. That is,*

$$\exists (\mathbf{X}, \mathbf{y}) \left[ \nexists f : \mathcal{R}(f \circ I, \mathbf{X}, \mathbf{y}) = \mathcal{R}^* \right].$$

Corollary 3.2 states that perfectly imputing missing data can reduce the best possible performance of a predictive model. This has substantial implications for how imputation is understood for prediction: if perfect imputation is achieved, then impute-then-predict models sacrifice expressiveness. If imputation is optimized to maintain the downstream performance of impute-then-predict models, the imputed data loses some of its meaning since it is no longer our "best guess" of the missing data's value, as we need to deliberately avoid perfect imputation to guarantee that we maintain performance.

Motivated by Proposition 3.1 and Corollary 3.2, we are interested in constructing predictive models that explicitly use missingness as a value in their prediction rather than imputing first. More generally, we may also consider using the indicator for each *type* of missing data directly in a prediction. Generalized additive models (GAMs) provide a natural choice for such a model.

A GAM $g : \mathbb{R}^d \to \mathbb{R}$ consists of a bias term $\beta_0 \in \mathbb{R}$ and $d$ shape functions $f_1, \ldots, f_d : \mathbb{R} \to \mathbb{R}$ parameterized by vectors $\beta_1, \ldots, \beta_d$. Given a sample $\mathbf{x}_i$, a GAM forms a prediction as:

$$g(\mathbf{x}_i; \beta) = \beta_0 + \sum_{j=1}^{d} f_j(x_{i,j}; \beta_j). \tag{1}$$

In practice, it is common for each shape function to be a linear combination of different thresholds on its input variable, i.e., $f_j(x_{i,j}; \beta_j) = \sum_{k=1}^{\text{len}(\mathbf{t}_j)} \beta_{j,k} \mathbf{1}_{[x_{i,j} \leq t_{j,k}]}$, where $\text{len}(\mathbf{t}_j) \in \mathbb{N}$ is the number of thresholds applied to variable $j$, each $t_{j,k} \in \mathbb{R}$ is a threshold value, and each $\beta_{j,k} \in \mathbb{R}$ is a learned weight.

These functions provide a convenient framework for considering missing values. In particular, we can form a new shape function $h_j(x_{i,j}; \beta_j, \beta_j^{\text{missing}})$ that explicitly handles missing data by introducing additional "missingness indicator" terms, such that our shape functions take the form

$$h_j(x_{i,j}; \beta_j, \beta_j^{\text{miss}}) = f_j(x_{i,j}; \beta_j) + \sum_{m=1}^{c} \beta_{j,m}^{\text{miss}} \mathbf{1}_{[mcat(x_{i,j})=m]},$$

where $\beta_j^{\text{miss}} \in \mathbb{R}^c$ is an additional vector of parameters. Recall that there are $c$ distinct reasons for missingness, with $mcat(x_{i,j}) = 0$ if $x_{i,j}$ is not missing and $mcat(x_{i,j}) = m$ if $x_{i,j}$ is missing for the $m$-th reason.

We may further extend this augmentation to include *interaction terms* between *missingness indicators and standard threshold functions*. The "missingness interaction" function between feature $j$ and feature $j'$ takes the form

$$h_{j,j'}(x_{i,j}, x_{i,j'}; \alpha_{j,j'}) = \sum_{m=1}^{c} \sum_{k=1}^{\text{len}(\mathbf{t}_j)} \alpha_{j,j',k,m} \mathbf{1}_{[mcat(x_{i,j})=m \text{ and } x_{i,j'} \leq t_{j',k}]},$$

where each $\alpha_{j,j',k,m} \in \mathbb{R}$ is a learned weight. We thus define a missingness-GAM (M-GAM) $g_{\text{miss}}$ as follows:

**Definition 3.3.** Given parameters $\alpha$, $\beta^{\text{miss}}$, and $\beta$, an M-GAM is defined as

$$g_{\text{miss}}(\mathbf{x}_i; \beta, \beta^{\text{miss}}, \alpha) = \beta_0 + \sum_{j=1}^{d} h_j(x_{i,j}; \beta_j, \beta_j^{\text{miss}}) + \sum_{j=1}^{d} \sum_{j'=1}^{d} h_{j,j'}(x_{i,j}, x_{i,j'}; \alpha_{j,j'}), \tag{2}$$

where

$$h_{j,j'}(x_{i,j}, x_{i,j'}; \alpha_{j,j'}) = \sum_{m=1}^{c} \sum_{k=1}^{\text{len}(\mathbf{t}_j)} \alpha_{j,j',k,m} \mathbf{1}_{[mcat(x_{i,j})=m \text{ and } x_{i,j'} \leq t_{j',k}]}$$

and

$$h_j(x_{i,j}; \beta_j, \beta_j^{\text{miss}}) = f_j(x_{i,j}; \beta_j) + \sum_{m=1}^{c} \beta_{j,m}^{\text{miss}} \mathbf{1}_{[mcat(x_{i,j})=m]}.$$

These augmentation terms are quite powerful. Theorem 3.4 shows that, for any impute-then-predict approach using an affine imputer and a GAM predictor, we can construct an M-GAM that recovers the expected classification score over imputations.

**Theorem 3.4.** *Consider any GAM $g : \mathbb{R}^d \to \mathbb{R}$, parameterized by $\beta$, with shape functions defined as linear combinations over boolean features (either thresholds $f_j(x_{i,j}; \beta_j) = \sum_{k=1}^{len(\mathbf{t}_j)} \beta_{j,k} \mathbf{1}_{[x_{i,j} \leq t_{j,k}]}$ or a feature that was originally boolean). Suppose some observations are missing boolean feature $b$, and that this feature is imputed such that the modeled probability of $x_{i,b}$ being true, $\hat{\mathbb{P}}(x_{i,b} = 1 | \mathbf{x}_{i,-b})$ (where $\mathbf{x}_{i,-b}$ refers to all covariates except $b$) is an affine function $h : \mathbf{x}_{i,-b} \to [0, 1]$. For any parameterization $\beta$ of a GAM $g$, let $\mathbb{E}[g(\mathbf{x}_i; \beta)] := \hat{\mathbb{P}}(x_{i,b} = 1 | \mathbf{x}_{i,-b}) g(\mathbf{x}_i^{(b+)}; \beta) + \hat{\mathbb{P}}(x_{i,b} = 0 | \mathbf{x}_{i,-b}) g(\mathbf{x}_i^{(b-)}; \beta)$, where $\mathbf{x}_i^{(b+)}$ denotes $\mathbf{x}_i$ with $x_{i,b} = 1$ and $\mathbf{x}_i^{(b-)}$ denotes $\mathbf{x}_i$ with $x_{i,b} = 0$. Then, there exists a model in the model class M-GAM (which does not use imputations), that recovers this score $\mathbb{E}[g(\mathbf{x}_i; \beta)]$ for all $i$.*

More broadly, Theorem 3.4 suggests that M-GAM is able to express scores comparable to those of any impute-then-predict GAM, if the imputation probabilities can be approximated by an additive model. One advantage is that M-GAM can be optimized directly for classification performance – rather than first optimizing an imputation step to recover missing values, and then optimizing a model on the imputed data. Together, Proposition 3.1 and Theorem 3.4 show that M-GAM is comparable to impute-then-predict in a broad range of settings and that M-GAM is strictly better than impute-then-predict in some settings. Appendix C contains the proof for Theorem 3.4.

## 3.1 Sparsity

Building a GAM with missingness indicators and interaction terms provides superior expressive power but causes an explosion of the number of covariates the model must consider. The GAM in Equation (1) consists of $\sum_j^d len(\mathbf{t}_j) + 1$ coefficients, while the M-GAM in Definition (3.3) consists of $c \sum_j^d \left( \sum_{j' \neq j} len(\mathbf{t}_{j'}) \right) + \sum_j^d (len(\mathbf{t}_j)) + cd + 1$ coefficients (or, $d \sum_j^d (len(\mathbf{t}_j)) + d + 1$ when $c = 1$). This increases the risk of overfitting and may lead to complex, uninterpretable models. The same problem arises when adding similar interaction terms to a linear model, as diagnosed by Van Ness et al. (2023), who propose a hypothesis testing style framework for variable selection. Notably, this framework does not explicitly encourage sparsity – it only discourages overfitting.

Rather than applying a pre-processing step for variable selection, we use $\ell_0$ regularization, which we can optimize directly alongside accuracy. This encourages the model coefficients to be 0, resulting in sparse models despite the potentially large number of input features. We optimize classification performance using the exponential loss, as it yields faster convergence rates than logistic loss during optimization (Liu et al., 2022). Thus, our goal is to solve the following optimization problem:

$$\min_{\beta, \beta^{\text{miss}}, \alpha} \left( \frac{1}{n} \sum_{i=1}^n e^{-y_i g_{\text{miss}}(\mathbf{x}_i; \beta, \beta^{\text{miss}}, \alpha)} + \lambda_0 (\|\beta\|_0 + \|\beta^{\text{miss}}\|_0 + \|\alpha\|_0) \right), \tag{3}$$

where $\lambda_0$ is a hyperparameter that determines the strength of the $\ell_0$ regularization.

To simplify (3) so it can be solved directly, we construct a new set of features $\bar{\mathbf{X}} \in \{0, 1\}^{c \sum_j^d \left( \sum_{j' \neq j} len(\mathbf{t}_{j'}) \right) + \sum_j^d (len(\mathbf{t}_j)) + cd + 1}$ consisting of the indicator for each threshold, missing value, and interaction term for each feature in the original dataset. For a large coefficient vector $\gamma \in \mathbb{R}^{c \sum_j^d \left( \sum_{j' \neq j} len(\mathbf{t}_{j'}) \right) + \sum_j^d (len(\mathbf{t}_j)) + cd + 1}$ and bias coefficient $\gamma_0 \in \mathbb{R}$, our optimization problem becomes:

$$\min_{\gamma} \frac{1}{n} \sum_{i=1}^n e^{-y_i (\gamma^T \bar{\mathbf{x}}_i + \gamma_0)} + \lambda_0 \|\gamma\|_0, \tag{4}$$

which is solved using the optimization framework of Liu et al. (2022). This allows us to quickly produce sparse M-GAMs, overcoming the large number of input values.

## 4 Experiments

We now evaluate the performance, runtime, and sparsity of M-GAM in comparison to other methods. To evaluate M-GAM in a realistic setting, we require datasets with some missing entries. We primarily

consider four datasets: the Explainable Machine Learning Challenge dataset (FICO et al., 2018) (referred to as FICO), a breast cancer dataset introduced by Razavi et al. (2018) (referred to as Breast Cancer), the MIMIC-III critical care dataset (Johnson et al., 2016) (referred to as MIMIC), and a dataset concerning the prediction of pharyngitis introduced by Miyagi (2023) (referred to as Pharyngitis). FICO contains 10,459 individuals, measuring 23 predictor variables used to predict whether each individual will repay a line of credit within 2 years. FICO contains three distinct encodings for missingness: -7, indicating no information of a given type is available, -8, indicating there was no usable information, and -9, indicating that a credit bureau report was not investigated or not found. Breast Cancer measures 27 features for 1,756 patients, MIMIC measures 49 features for 30,238 patients, and Pharyngitis measures 19 features for 676 patients. We use AUC, rather than accuracy, when evaluating model performance for Breast Cancer and MIMIC because these two datasets are heavily imbalanced. Breast Cancer, MIMIC, and Pharyngitis contain only one missingness encoding. Two additional datasets are studied in Section E of the appendix.

Each dataset contains missing entries. Because these are real datasets, we do not know the exact mechanism(s) (i.e., MCAR, MAR, or MNAR) by which data are missing. These datasets allow us to evaluate M-GAM on data with (Section 4.1) and without (Section 4.4) added MAR missingness.

We then study the interpretability/accuracy tradeoff for M-GAM using sparsity versus accuracy plots (Section 4.2) and evaluate the runtime of M-GAM (Section 4.3). We use multivariate imputation by chained equations (MICE) (Van Buuren & Oudshoorn, 1999), MIWAE(Mattei & Frellsen, 2019), and MissForest(Stekhoven & Bühlmann, 2012) as multiple imputation baselines.

We compare M-GAM to a variety of standard machine learning models used in an impute-then-predict framework. We further compare to standard machine learning models used with both the missingness augmentation described by Van Ness et al. (2023) and imputation. Section D.3 of the appendix contains full experimental details.

## 4.1 M-GAM Provides Superior Performance Given Informative MAR Missingness

To demonstrate the added expressive capability of our model relative to impute-then-predict models, we created versions of each dataset with added synthetic missingness. Missingness is added to an arbitrary column of the data according to an MAR mechanism, where missingness is dependent on the outcome $Y$ and one other randomly chosen predictor variable. We encode this added missingness as a new value distinct from the value(s) used to indicate missingness in the original dataset. A conditional probability table for this synthetic missingness is provided in Appendix Section D.1.

This adjustment falls under the MAR setting where imputation is often suggested. Nevertheless, as shown in Figure 3, M-GAM with interactions provides much greater accuracy than imputation, because the missingness depends on the outcome. The gain in performance due to considering interaction terms grows larger with increasing MAR missingness (from left to right).

Note that missingness depending on the outcome of interest is realistic. For example, individuals who are unlikely to have a particular disease are unlikely to receive medical tests related to that disease. One theory about why polls were wrong before the 2016 US presidential election was that non-response bias was associated with less education and distrust in the media, both predictors of votes for Donald Trump (Kurtzleben, 2016).

## 4.2 M-GAM Achieves High Performance While Maintaining Sparsity

The most important benefit of the missingness handling in M-GAM is that it enables simple, sparse models. As such, we show the tradeoff between complexity and performance for M-GAM.

Figure 4 demonstrates the sparsity-accuracy trade-off for M-GAM relative to a GAM fit using Scikit Learn's logistic regression package (Pedregosa et al., 2011) over binarized features, trained on data from 10 imputations using a multiple imputation method. We also contrast two different levels of missing variable parameterization: the full set of indicators and interactions versus using just indicators.

We quantify interpretability using the number of nonzero coefficients selected by M-GAM, since a large number of non-zero coefficients leads to a dense, complicated mapping from the input data to a prediction. Meanwhile, running impute-then-predict is not interpretable: the method

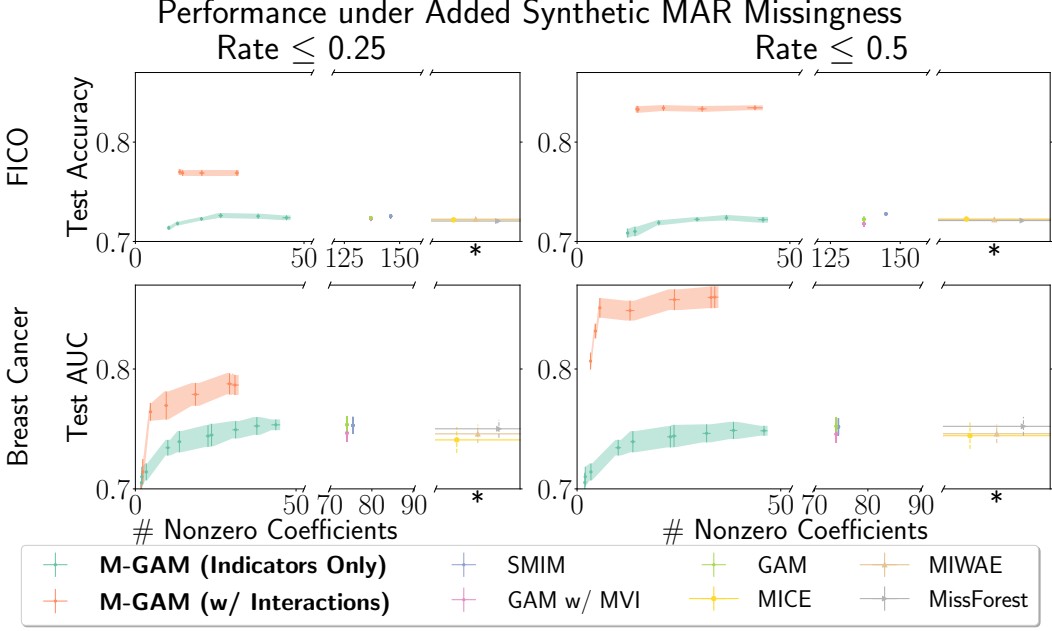

Figure 3: Sparsity of M-GAM when synthetic MAR missingness is added to up to 25% (left column) and 50% (right column) of entries in FICO (top row) and Breast Cancer (bottom row). We compare to several alternatives for GAMs with missing data: ensembling 10 GAMs fit on multiple imputation (for MIWAE, MICE, and MissForest), 0-value imputation ("GAM"), mean-value imputation ("GAM w/ MVI"), and selective addition of missingness indicators ("SMIM"). The number of non-zero coefficients for multiple imputation cannot be evaluated because the models depend on both the GAM coefficients and the underlying imputation mechanisms, resulting in high dimensional shape functions as in Figure 1. Error bars report standard error over 10 train-test splits.

requires ensembling many different GAMs, and the imputations themselves introduce complicated relationships between the raw data and the classifications, similar to what was illustrated in Figure 1. We show that with fewer than 40 total coefficients (including all step functions for all variables), M-GAM can achieve accuracy comparable to that of GAMs with multiple imputation. On FICO, M-GAM – using just 20 non-zero coefficients – achieves superior accuracy to a variety of dense, complicated alternatives.

### 4.3   M-GAM is Faster than Impute-then-Predict

We next turn to a runtime comparison between the impute-then-predict framework and M-GAM. For impute-then-predict models, we first imputed 10 datasets and recorded the time required to do so. We then fit a predictive model on each imputed training dataset and recorded the total time required. We recorded the time required to fit an M-GAM with missingness indicators and an M-GAM with missingness interactions for comparison. This was repeated for each of ten distinct train-test splits of the original dataset.

Figure 5 shows the runtime of our approach relative to each impute-then-predict baseline, as well as decision trees and random forests without imputation. M-GAM consistently produces models at least an order of magnitude more quickly than impute-then-predict with any non-trivial imputation. While decision trees without imputation tend to be produced faster than M-GAM, they tend to have lower accuracy than M-GAM as discussed in the next section. We repeat this experiment for four distinct subsamples of each dataset (1/4, 1/2, 3/4, and all of the data) to study how each method scales in the number of samples in Appendix E.3.

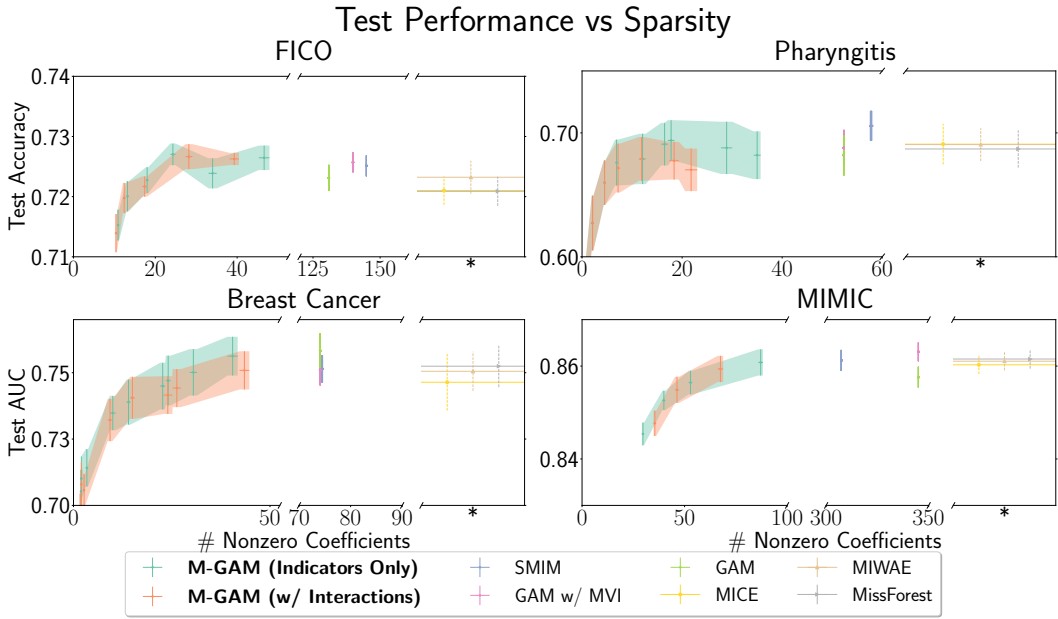

Figure 4: Test performance of three models at various levels of sparsity on the unaltered FICO and Breast Cancer datasets, with the same baselines as in Figure 3

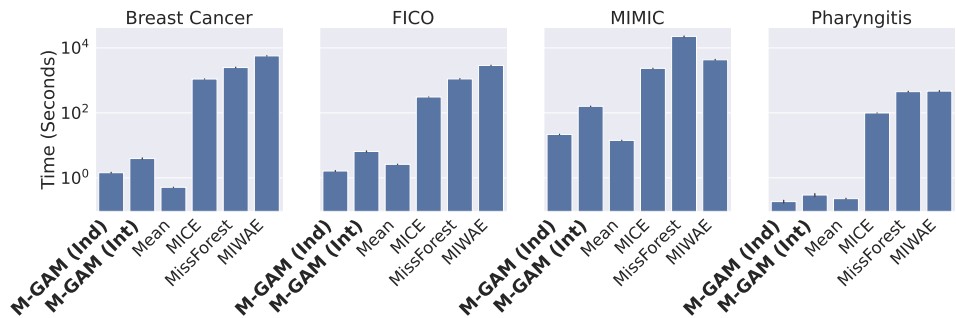

Figure 5: Runtime of different methods on Breast Cancer, FICO, MIMIC, and Pharyngitis. For each imputation method, we report the total time required to impute missing data and fit the best performing impute-then-predict classifier for that dataset and imputation method. M-GAM (Ind) is an M-GAM with indicators and M-GAM (Int) is an M-GAM with indicators and interaction terms. Error bars report standard error of total runtime over 10 train-test splits.

### 4.4 M-GAM is as Accurate as Impute-then-Predict on Real Data

While M-GAM outperforms imputation on semi-synthetic data, there is a risk that this comes at the cost of performance on real data. To evaluate whether this is the case, we used several multiple imputation methods to impute 10 distinct datasets for each setting, then fit a variety of predictive models on these datasets. We used cross validation to select hyperparameters separately for each imputed training dataset and ensembled the resulting 10 models for each model class to produce a single predictive model for each model class. We repeated this procedure for ten distinct train-test splits for each dataset considered.

Figure 6 shows the test accuracy of each model. We find that, on real datasets, no alternative method substantially outperforms M-GAM. This suggests that M-GAM does not harm predictive performance on real datasets, while providing substantial benefits in interpretability (Section 4.2) and superior power under informative missingness (Section 4.1).

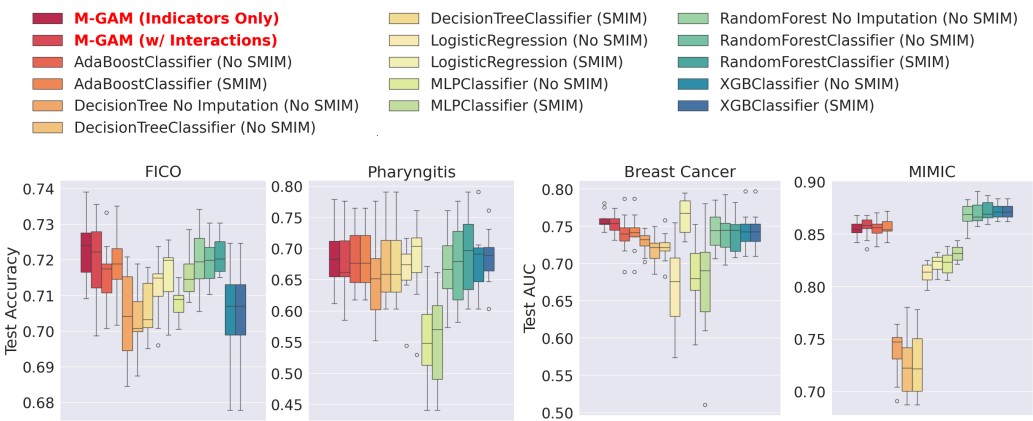

Figure 6: Box-and-whiskers plots comparing test performance of baseline models to M-GAM on four datasets over ten train-test splits. All methods except M-GAMs, "DecisionTree No Imputation", and "RandomForest No Imputation" impute ten datasets using MICE then ensemble the models fit on each dataset.

## 5 Conclusion

We introduced M-GAM, a framework for producing accurate, sparse GAMs in the presence of missing data. We demonstrated that M-GAM achieves comparable accuracy to impute-then-predict on real datasets and superior accuracy under informative synthetic missingness. M-GAM produces models substantially more quickly than impute-then-predict models with multiple imputation, and provides simple, transparent reasoning on missing data.

While the $\ell_0$ penalty in M-GAM encourages sparsity, it is limited in that the $\ell_0$ regularization is applied uniformly across all coefficients. Consider the case when we are adding interaction terms to handle missingness. We might encourage the model to rely on features it is already using to predict y, rather than using new features. It may be more effective regularization – and more interpretable – to have a reduced $\ell_0$ penalty for such cases. Future work should investigate applying distinct levels of regularization to observed variables, missingness indicators, and missingness interactions when a variable is already included in the model.

An important caveat to models that reason on missing features is that missingness can be especially vulnerable to distribution shift, particularly in a medical domain (Sperrin et al., 2020; Groenwold, 2020). The interpretability enabled by M-GAM is crucial in allowing models to be closely monitored and adjusted in the presence of potential distribution shift. Future work could more thoroughly investigate potential distribution shift and ways to adjust a model which reasons on missing data.

On the whole, M-GAM quickly produces accurate, interpretable models, providing a new degree of transparency to predictions in the presence of missing data. The code used for this work is available at https://github.com/jdonnelly36/M-GAM.

**Societal Impacts.** M-GAM offers an interpretable way to deploy machine learning in high stakes domains like medicine, even when data is missing. Modeling decisions for missing data risk introducing or perpetuating unfairness Jeanselme et al. (2022). We view interpretability as a key tool for addressing this.

## 6 Acknowledgements

We acknowledge funding from the National Institutes of Health under 5R01-DA054994, the National Science Foundation under grant HRD-2222336 and the Department of Energy under grant DE-SC002135. Additionally, this material is based upon work supported by the National Science Foundation Graduate Research Fellowship under Grant No. DGE 2139754. Finally, we thank Jiachang Liu for his helpful advice.

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

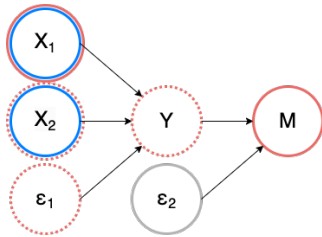

Figure 7: The case constructed to prove Proposition A.1. Model $f_1$ takes as input variables in blue circles, $f_2$ takes as input variables in red circles, and dashed red circles indicate variables involved in generating missingness in $X_1$, denoted by $M$. Here, $\epsilon_1$ and $\epsilon_2$ are unmeasured noise. By using $M$ as an input, $f_2$ can infer information about $Y$ *after* noise from $\epsilon_1$ is considered.

# A    Proof of Proposition 3.1

First, recall Proposition 3.1:

**Proposition A.1.** *Let $I : \mathbb{R}^d \to \mathbb{R}^d$ be an oracle imputation function that replaces all missing values in a vector with the correct non-missing entry. For a random variable $X \in \mathbb{R}^d$, let $f_1(X) := \mathbf{1}_{[\mathbb{E}[Y|I(X)]>0.5]}$ be the Bayes' optimal model using perfectly imputed data and $f_2(X) := \mathbf{1}_{[\mathbb{E}[Y|X]>0.5]}$ be the Bayes' optimal model using missingness as a value. There exist data generating processes for $X$ and $Y$ where $P(Y = f_1(X)) < P(Y = f_2(X))$.*

*Proof.* We prove Proposition 3.1 by construction.

Let $X_1, X_2 \sim$ Bernoulli$(p = 0.5)$ and $Y := |X_1 X_2 - \epsilon_1|$ where $\epsilon_1 \sim$ Bernoulli$(p = k_1)$ is unobserved noise. Let $M$ denote missingness in $X_1$, where $M := |Y - \epsilon_2|$ with $\epsilon_2 \sim$ Bernoulli$(p = k_2)$ being unobserved noise. Let $k_2 < k_1 < 0.5$.

Consider two oracle models $f_1$ and $f_2$, defined as:

$$f_1(X_1, X_2) := \mathbf{1}_{[\mathbb{E}[Y|I(X)]\geq0.5]} = \mathbf{1}_{[\mathbb{E}[Y|X_1,X_2]\geq0.5]} \quad f_2(M, X_2) := \mathbf{1}_{[\mathbb{E}[Y|X]\geq0.5]} = \mathbf{1}_{[\mathbb{E}[Y|M,X_2]\geq0.5]}.$$

Intuitively, $f_1$ perfectly imputes $X_1$ when $X_1$ is missing, then predicts $Y$ using $X_1$ and $X_2$, while $f_2$ predicts $Y$ using $M$ and $X_2$.

We first evaluate the expected accuracy of the imputation model, i.e., $\mathbb{P}(f_1(X_1, X_2) = Y)$. Using the law of total probability, we have:

$$
\begin{aligned}
\mathbb{P}(f_1(X_1, X_2) = Y) &= \sum_{x_1,x_2} \mathbb{P}(x_1, x_2)\mathbb{P}(f_1(x_1, x_2) = Y|x_1, x_2) \\
&= \sum_{x_1,x_2} \mathbb{P}(x_1)\mathbb{P}(x_2)\mathbb{P}(f_1(x_1, x_2) = Y|x_1, x_2) \qquad \text{Since } X_1 \perp X_2 \\
&= 0.25 \sum_{x_1,x_2} \mathbb{P}(f_1(x_1, x_2) = Y|x_1, x_2)
\end{aligned}
$$

Noting that, when $X_1 = X_2 = 1$, we have $Y = |1 - \epsilon_1|$, so $\mathbb{P}(Y = 1|X_1 = 1, X_2 = 1) = \mathbb{P}(\epsilon_1 = 0) = 1 - k_1$. When at least one of $X_1$ and $X_2$ is 0, we have $Y = |0 - \epsilon_1|$, so $\mathbb{P}(Y = 1|X_1 X_2 = 0) = \mathbb{P}(\epsilon_1 = 1) = k_1$. Since $f_1$ simply evaluates whether the expectation of $Y$ given $X_1$ and $X_2$ is greater than 0.5, if $k_1 < 0.5$, we have $f_1(1, 1) = 1$ and $f_1(0, 1) = f_1(1, 0) = f_1(0, 0) = 0$. Thus, the expected accuracy is:

$$
\begin{aligned}
\mathbb{P}(f_1(X_1, X_2) = Y) &= 0.25 \sum_{x_1,x_2} \mathbb{P}(f_1(x_1, x_2) = Y|x_1, x_2) \\
&= 0.25 \sum_{x_1,x_2} \mathbb{P}(\epsilon_1 = 0) \\
&= \mathbb{P}(\epsilon_1 = 0) \\
&= 1 - k_1
\end{aligned}
$$

Similarly,

$$\mathbb{P}(f_2(m, x_2) = Y) = \sum_{m, x_2} \mathbb{P}(m, x_2)\mathbb{P}(f_2(m, x_2) = Y|m, x_2)$$

$$= \sum_{m, x_2} \mathbb{P}(x)\mathbb{P}(m|x_2)\mathbb{P}(f_2(m, x_2) = Y|m, x_2)$$

$$= 0.5\big(\mathbb{P}(m = 1|x_2 = 1)\mathbb{P}(f_2(m, x_2) = Y|m, x_2) + \mathbb{P}(m = 1|x_2 = 0)\mathbb{P}(f_2(m, x_2) = Y|m, x_2)$$

$$+ \mathbb{P}(m = 0|x = 1)\mathbb{P}(f_2(m, x_2) = Y|m, x_2) + \mathbb{P}(m = 0|x_2 = 0)\mathbb{P}(f_2(m, x_2) = Y|m, x_2)\big)$$

We can compute each term in the conditional probability table $P(M|X_2)$ as follow:

$$\mathbb{P}(m = 0|x_2 = 0) = \sum_y \mathbb{P}(m = 0|x_2 = 0, y)\mathbb{P}(y|x_2 = 0)$$

$$= (1 - k_2)(1 - k_1) + k_2 k_1$$

$$\mathbb{P}(m = 1|x_2 = 0) = \sum_y \mathbb{P}(m = 1|x_2 = 0, y)\mathbb{P}(y|x_2 = 0)$$

$$= k_2(1 - k_1) + (1 - k_2)k_1$$

$$\mathbb{P}(m = 0|x_2 = 1) = \sum_y \mathbb{P}(m = 0|x_2 = 1, y)\mathbb{P}(y|x_2 = 1)$$

$$= k_2(0.5k_1 + 0.5(1 - k1)) + (1 - k_2)(0.5(1 - k_1) + 0.5k1)$$

$$= 0.5$$

$$\mathbb{P}(m = 1|x_2 = 1) = \sum_y \mathbb{P}(m = 1|x_2 = 1, y)\mathbb{P}(y|x_2 = 1)$$

$$= (1 - k_2)(0.5k_1 + 0.5(1 - k_1)) + k_2(0.5(1 - k_1) + 0.5k_1)$$

$$= 0.5$$

Plugging these values in, we have

$$\mathbb{P}(f_2(M, X_2) = Y) = 0.5\bigg((1 - k_1 - k_2 + 2k_1 k_2)\mathbb{P}(f_2(m, x_2) = Y|m = 0, x_2 = 0)$$

$$+ (k_2 + k_1 - 2k_1 k_2)\mathbb{P}(f_2(m, x_2) = Y|m = 1, x_2 = 0)$$

$$+ 0.5\mathbb{P}(f_2(m, x_2) = Y|m = 0, x_2 = 1)$$

$$+ 0.5\mathbb{P}(f_2(m, x_2) = Y|m = 1, x_2 = 1)\bigg)$$

$$\geq 0.5\bigg((1 - k_1 - k_2 + 2k_1 k_2)\mathbb{P}(f_2(m, x_2) = Y|m = 0)$$

$$+ (k_2 + k_1 - 2k_1 k_2)\mathbb{P}(f_2(m, x_2) = Y|m = 1)$$

$$+ 0.5\mathbb{P}(f_2(m, x_2) = Y|m = 0)$$

$$+ 0.5\mathbb{P}(f_2(m, x_2) = Y|m = 1)\bigg)$$

$$= 0.5(1 - k_1 - k_2 + 2k_1 k_2 + k_2 + k_1 - 2k_1 k_2 + 0.5 + 0.5)(1 - k_2)$$

$$= 1 - k_2$$

Thus, we have $\mathbb{P}(f_1(X_1, X_2) = Y) = 1 - k_1$ and $\mathbb{P}(f_2(M, X_2) = Y) \geq 1 - k_2$. Since $k_2 < k_1$, we have

$$k_2 < k_1$$

$$1 - k_2 > 1 - k_1$$

$$\mathbb{P}(f_2(M, X_2) = Y) > \mathbb{P}(f_1(X_1, X_2) = Y),$$

as required, where the last step follows because $\mathbb{P}(f_1(X_1, X_2) = Y) \geq 1 - k_2$. $\qquad\square$

## A.1 Alternative Proof

While the above offers a proof of Proposition 3.1, one might wonder whether the proposition holds outside of the case we constructed. In the previous proof, we aimed for a case that resulted in an easy to follow proof. Here, we prove Proposition 3.1 by a second construction, to show that imputation can be worse than missingness-as-a-value even if there is more noise in the informative missingness than in the data itself, and even when missingness is relatively rare.

*Proof.* Consider a case similar to that used for the proof in proposition 3.1, where we add an additional variable $X_3$ and have the noise for the missingness be higher than any other noise.

$$Y = |X_1 X_2 - \epsilon_1|, \epsilon_1 \sim \text{Bern}(\frac{1}{12})$$

$$M = \begin{cases} |Y - \epsilon_2|, \epsilon_2 \sim \text{Bern}(\frac{1}{4}), & \text{with probability } \frac{1}{2} \\ 0, & \text{with probability } \frac{1}{2} \end{cases}$$

$$X_3 = |Y - \epsilon_3|, \epsilon_3 \sim \text{Bern}(\frac{1}{11})$$

We also adjust the probabilities for $X_1$ and $X_2$ being true so that this is a balanced classification problem: $X_1, X_2 \sim \text{Bern}(\frac{1}{\sqrt{2}})$, so $X_1 X_2 \sim \text{Bern}(\frac{1}{2})$,

Note that we now also have missingness at well under 50% of the data (missingness happens a quarter of the time).

The bayes optimal model with perfect imputation of $X_1$, and no access to $M$, is still just to predict in accordance with $X_1 X_2$ for $\mathbb{P}(X_1 X_2 = Y) = \frac{11}{12}$. When $X_1 X_2 = X_3$, all information available suggests $X_1 X_2$ is correct. When $X_1 X_2 \neq X_3$, we still have the bayes optimal prediction aligning with $X_1 X_2$: $\mathbb{P}(Y = 1 | X_1 X_2 = 1, X_3 = 0) = \frac{11}{21} > 0.5$ and $\mathbb{P}(Y = 0 | X_1 X_2 = 0, X_3 = 1) = \frac{11}{21} > 0.5$.

If we instead only have access to $X_1$ when it is not missing, but we also know when $X_1$ is missing (i.e. we know $M$), then the following approach will perform better than the above model:

$$Y = \begin{cases} X_3, & \text{if } M = 1 \\ X_1 X_2 X_3, & \text{if } M = 0 \end{cases}$$

When $M = 1$, we make additional errors relative to the previous approach at rate

$$\mathbb{P}(M = 1)(\mathbb{P}((X_3) \neq Y) - \mathbb{P}((X_1 X_2) \neq Y))$$
$$= \frac{1}{4}(\frac{1}{11} - \frac{1}{12})$$
$$= \frac{1}{528}$$

When $M = 0$, we improve our classifier's accuracy by:

$$\mathbb{P}(\text{Imputation model is wrong, model with missingness is right and } M = 0)$$
$$- \mathbb{P}(\text{Imputation model is right, model with missingness is wrong and } M = 0)$$
$$= \mathbb{P}(X_1 X_2 \neq Y = X_1 X_2 X_3, M = 0) - \mathbb{P}(X_1 X_2 = Y \neq X_1 X_2 X_3, M = 0)$$
$$= \mathbb{P}(X_1 X_2 = 1, X_3 = 0, Y = 0, M = 0) - \mathbb{P}(X_1 X_2 = 1, X_3 = 0, Y = 1, M = 0)$$
$$= \mathbb{P}(X_1 X_2 = 1)\mathbb{P}(Y = 0 | X_1 X_2 = 1)\mathbb{P}(X_3 = 0 | Y = 0)\mathbb{P}(M = 0 | Y = 0)$$
$$- \mathbb{P}(X_1 X_2 = 1)\mathbb{P}(Y = 1 | X_1 X_2 = 1)\mathbb{P}(X_3 = 0 | Y = 1)\mathbb{P}(M = 0 | Y = 1)$$
$$= \frac{1}{2}\frac{1}{12}\frac{10}{11}\frac{7}{8} - \frac{1}{2}\frac{11}{12}\frac{1}{11}\frac{5}{8}$$
$$= \frac{15}{2112}$$
$$> \frac{1}{528}$$

That is, the proportion of cases the classifier that uses missingness will gain is greater than the proportion it will lose relative to the imputation approach. So, the model that uses missingness outperforms the imputation model. □

# B  Proof of Corollary 3.2

**Corollary B.1.** *Let $\mathcal{R}(f, \mathbf{X}, \mathbf{Y})$ denote the risk of a model $f$ for data $\mathbf{X}, \mathbf{Y}$, and $\mathcal{R}^*$ the optimal risk. Let $I : (\mathbb{R} \cup NA)^d \to \mathbb{R}^d$ denote the oracle imputation function of Proposition 3.1. Under perfect imputation, it is possible for there to be no Bayes optimal model built on imputed data. That is,*

$$\exists(\mathbf{X}, \mathbf{Y}) \left[ \nexists f : \mathcal{R}(f \circ I, \mathbf{X}, \mathbf{Y}) = \mathcal{R}^* \right].$$

*Proof.* Recall that, in Proposition A.1, we constructed a distribution of $X, Y$ such that

$$\mathcal{R}(f_1, X, Y) > \mathcal{R}(f_2, X, Y)$$

where

$$f_1 = \arg\min_f \mathcal{R}(f, I(X), Y)$$
$$f_2 = \arg\min_f \mathcal{R}(f, X, Y)$$

By definition of $\mathcal{R}^*$, we have that $\mathcal{R}(f_2, X, Y) \geq \mathcal{R}^*$, immediately yielding

$$\mathcal{R}(f_1, X, Y) > \mathcal{R}(f_2, X, Y) \geq \mathcal{R}^*.$$

$\square$

# C Proof of Theorem 3.4

Theorem 3.4 states:

**Theorem C.1.** *Consider any GAM $g : \mathbb{R}^d \to \mathbb{R}$, parameterized by $\beta$, with shape functions defined as linear combinations over boolean features (either thresholds $f_j(x_{i,j}; \beta_j) = \sum_{k=1}^{len(\mathbf{t}_j)} \beta_{j,k} \mathbf{1}_{[x_{i,j} \leq t_{j,k}]}$ or a feature that was originally boolean). Suppose some observations are missing boolean feature $b$, and that this feature is imputed such that the modeled probability of $x_{i,b}$ being true, $\hat{\mathbb{P}}(x_{i,b} = 1 | \mathbf{x}_{i,-b})$ (where $\mathbf{x}_{i,-b}$ refers to all covariates except $b$) is an affine function $h : \mathbf{x}_{i,-b} \to [0, 1]$. For any parameterization $\beta$ of $g$, let $\mathbb{E}[g(\mathbf{x}_i; \beta)] := \hat{\mathbb{P}}(x_{i,b} = 1 | \mathbf{x}_{i,-b}) g(\mathbf{x}_i^{(b+)}; \beta) + \hat{\mathbb{P}}(x_{i,b} = 0 | \mathbf{x}_{i,-b}) g(\mathbf{x}_i^{(b-)}; \beta)$, where $\mathbf{x}_i^{(b+)}$ denotes $\mathbf{x}_i$ with $x_{i,b} = 1$ and $\mathbf{x}_i^{(b-)}$ denotes $\mathbf{x}_i$ with $x_{i,b} = 0$. Then, there exists a model in the model class M-GAM (which does not use imputations), that recovers this score $\mathbb{E}[g(\mathbf{x}_i; \beta)]$ for all $i$.*

*Proof.* [2]

## C.1 Model Family Definition for $\mathbb{E}[g(\mathbf{x}_i; \beta)]$

The general GAM, as per equation (1), is $g(\mathbf{x}_i; \beta) = \beta_0 + \sum_{j=1}^{d} f_j(x_{i,j}; \beta_j)$, or with the shape functions incorporated,

$$g(\mathbf{x}_i; \beta) = \beta_0 + \sum_{j=1}^{d} \sum_{k=1}^{len(\mathbf{t}_j)} \beta_{j,k} \mathbf{1}_{[x_{i,j} \leq t_{j,k}]}$$

where $\mathbf{1}_{[x_{i,j} \leq t_{j,k}]}$ values corresponding to a missing feature $j$ are replaced with some imputed value based on a GAM. To highlight the sometimes-missing boolean feature $b$, we may rewrite this as

$$g(\mathbf{x}_i; \beta) = \beta_0 + \left( \sum_{j \neq b} \sum_{k=1}^{len(\mathbf{t}_j)} \beta_{j,k} \mathbf{1}_{[x_{i,j} \leq t_{j,k}]} \beta_b \mathbf{1}_{[x_{i,b}]} \right) + \beta_b \mathbf{1}_{[x_{i,b}]}$$

For examples $\mathbf{x}_i$ with feature $b$ missing, the value $\mathbf{1}_{[x_{i,b}]}$ must be imputed prior to being used. Under the setting of this proof, the feature $b$ is imputed by modeling the probability as an affine function of an additive score. Without loss of generality, that means there exists some additive model with score: $s(\mathbf{x}_{i,-b}) = C_0 + \sum_{j \neq b} \sum_{k=1}^{len(\mathbf{t}_j)} C_{j,k} \mathbf{1}_{[x_{i,j} \leq t_{j,k}]}$ for some set of real-valued coefficients $C$, with corresponding probability of $x_{i,b} = 1$ given, WLOG, by:

$$\hat{\mathbb{P}}(x_{i,b} = 1 | \mathbf{x}_{i,-b}) = a \frac{s(\mathbf{x}_{i,-b}) - \min_{i'}(s(\mathbf{x}_{i',-b}))}{\max_{i'}(s(\mathbf{x}_{i',-b})) - \min_{i'}(s(\mathbf{x}_{i',-b}))} + d$$

where $a > 0, d > 0, a + d = 1$. Note that we can set $C_0 = 0$ without loss of generality because it will appear in both terms in the numerator and both terms in the denominator, having no impact on the overall probability.

The values $\max_{i'}(s(\mathbf{x}_{i',-b}))$ and $\min_{i'}(s(\mathbf{x}_{i',-b}))$ denote, respectively, the maximum and minimum of the score function over any possible $\mathbf{x}_i$.

If we take the expectation of the score function over the probability distribution learned by the imputing additive model, we get:

$$g(\mathbf{x}_i; \beta) = \beta_0 + \left( \sum_{j' \neq b} \sum_{k=1}^{len(\mathbf{t}_j)} \beta_{j',k} \mathbf{1}_{[x_{i,j'} \leq t_{j',k}]} \right) + \beta_b \mathbf{1}_{[x_{i,b}]} + \beta_b \mathbf{1}_{[mcat(x_{i,b}) \neq 0]} \hat{\mathbb{P}}(x_{i,b} = 1 | \mathbf{x}_{i,-b})$$

---

[2] Note that it is sufficient to show that this this theorem holds for a single missingness reason, $m = 1$. Showing that this theorem holds for M-GAM in the less expressive case where there is only a single reason for missingness, also shows that the theorem holds when there is added expressiveness to the model class. As such, we do not include notation for distinct missingness reasons in the proof.

(Recalling that $\mathbf{1}_{[x_{i,b}]}$ is defined to always be false when $x_{i,b}$ is missing, and $\mathbf{1}_{mcat(x_{i,b})\neq 0}$ is true iff $x_{i,b}$ is missing.)

We can put the $\beta_b \mathbf{1}_{mcat(x_{i,b})\neq 0}$ term back in the summation over thresholds and have:

$$g(\mathbf{x}_i; \beta) = \beta_0 + \left( \sum_{j=1}^{d} \sum_{k=1}^{\text{len}(\mathbf{t}_j)} \beta_{j,k} \mathbf{1}_{[x_{i,j} \leq t_{j,k}]} \right) + \beta_b \mathbf{1}_{[mcat(x_{i,b})\neq 0]} \hat{\mathbb{P}}(x_{i,b} = 1 | \mathbf{x}_{i,-b})$$

Simplifying further:

$$g(\mathbf{x}_i; \beta) = \beta_0 + \left( \sum_{j=1}^{d} \sum_{k=1}^{\text{len}(\mathbf{t}_j)} \beta_{j,k} \mathbf{1}_{[x_{i,j} \leq t_{j,k}]} \right) + \beta_b \mathbf{1}_{[mcat(x_{i,b})\neq 0]} \left( a \frac{s(x_{-b}) - \min_{i'}(s(\mathbf{x}_{i',-b}))}{\max_{i'}(s(\mathbf{x}_{i',-b})) - \min_{i'}(s(\mathbf{x}_{i',-b}))} + d \right)$$

$$g(\mathbf{x}_i; \beta) = \beta_0 + \left( \sum_{j=1}^{d} \sum_{k=1}^{\text{len}(\mathbf{t}_j)} \beta_{j,k} \mathbf{1}_{[x_{i,j} \leq t_{j,k}]} \right) + \beta_b \mathbf{1}_{[mcat(x_{i,b})\neq 0]} a \frac{s(x_{-b}) - \min_{i'}(s(\mathbf{x}_{i',-b}))}{\max_{i'}(s(\mathbf{x}_{i',-b})) - \min_{i'}(s(\mathbf{x}_{i',-b}))}$$
$$+ \beta_b \mathbf{1}_{[mcat(x_{i,b})\neq 0]} d$$

$$g(\mathbf{x}_i; \beta) = \beta_0 + \left( \sum_{j=1}^{d} \sum_{k=1}^{\text{len}(\mathbf{t}_j)} \beta_{j,k} \mathbf{1}_{[x_{i,j} \leq t_{j,k}]} \right) + \beta_b a \frac{s(x_{-b})}{\max_{i'}(s(\mathbf{x}_{i',-b})) - \min_{i'}(s(\mathbf{x}_{i',-b}))}$$
$$- \beta_b \mathbf{1}_{[mcat(x_{i,b})\neq 0]} a \frac{\min_{i'}(s(\mathbf{x}_{i',-b}))}{\max_{i'}(s(\mathbf{x}_{i',-b})) - \min_{i'}(s(\mathbf{x}_{i',-b}))} + \beta_b \mathbf{1}_{[mcat(x_{i,b})\neq 0]} d$$

$$g(\mathbf{x}_i; \beta) = \beta_0 + \sum_{j=1}^{d} \sum_{k=1}^{\text{len}(\mathbf{t}_j)} \beta_{j,k} \mathbf{1}_{[x_{i,j} \leq t_{j,k}]}$$
$$+ s(x_{-b}) \mathbf{1}_{[mcat(x_{i,b})\neq 0]} \frac{\beta_b a}{\max_{i'}(s(\mathbf{x}_{i',-b})) - \min_{i'}(s(\mathbf{x}_{i',-b}))}$$
$$+ \mathbf{1}_{[mcat(x_{i,b})\neq 0]} \left( \beta_b d - \frac{\beta_b a \min_{i'}(s(\mathbf{x}_{i',-b}))}{\max_{i'}(s(\mathbf{x}_{i',-b})) - \min_{i'}(s(\mathbf{x}_{i',-b}))} \right)$$

We can return to our shape function notation and write:

$$g(\mathbf{x}_i; \beta) = \beta_0 + \sum_{j=1}^{d} f_j(x_{i,j}; \beta_j)$$
$$+ \mathbf{1}_{[mcat(x_{i,b})\neq 0]} s(x_{-b}) \frac{\beta_b a}{\max_{i'}(s(\mathbf{x}_{i',-b})) - \min_{i'}(s(\mathbf{x}_{i',-b}))} \tag{5}$$
$$+ \mathbf{1}_{[mcat(x_{i,b})\neq 0]} \left( \beta_b d - \frac{\beta_b a \min_{i'}(s(\mathbf{x}_{i',-b}))}{\max_{i'}(s(\mathbf{x}_{i',-b})) - \min_{i'}(s(\mathbf{x}_{i',-b}))} \right)$$

### C.2 M-GAM Model Family Definition

Recall from section 3, Equation (3.3), that the form of M-GAM is:

$$g_{\text{miss}}(\mathbf{x}_i; \bar{\beta}, \bar{\beta}^{\text{miss}}, \bar{\alpha}) = \bar{\beta}_0 + \sum_{j=1}^{d} h_j(x_{i,j}; \bar{\beta}_j, \bar{\beta}_j^{\text{miss}}) + \sum_{j=1}^{d} \sum_{j'=1}^{d} h_{j,j'}(x_{i,j}, x_{i,j'}; \bar{\alpha}_{j,j'}).$$

Filling in the shape functions, we have:

$$g_{\text{miss}}(\mathbf{x}_i; \bar{\beta}, \bar{\beta}^{\text{miss}}, \bar{\alpha}) = \bar{\beta}_0 + \sum_{j=1}^{d} \left( f_j(x_j, \bar{\beta}_j) + \sum_{m=1}^{c} \bar{\beta}_{j,m}^{\text{miss}} \mathbf{1}_{[mcat(x_{i,j})=m]} \right)$$

$$+ \sum_{j=1}^{d} \sum_{j'=1}^{d} \sum_{m=1}^{c} \sum_{k=1}^{\text{len}(\mathbf{t}_j)} \bar{\alpha}_{j,j',k,m} \mathbf{1}_{[mcat(x_{i,j})=m \text{ and } x_{i,j'} \leq t_{j',k}]}. \tag{6}$$

### C.3 Show that any linear imputation model has an equivalent representation as a missingness term model

To show this, it is sufficient to show that for any coefficients of a linear imputation model, there exists a parameterization of M-GAM such that Equation (6) is equivalent to Equation (5).

Start from Equation (6):

$$g_{\text{miss}}(\mathbf{x}_i; \bar{\beta}, \bar{\beta}^{\text{miss}}, \bar{\alpha}) = \bar{\beta}_0 + \sum_{j=1}^{d} \left( f_j(x_j, \bar{\beta}_j) + \sum_{m=1}^{c} \bar{\beta}_{j,m}^{\text{miss}} \mathbf{1}_{[mcat(x_{i,j})=m]} \right)$$

$$+ \sum_{j=1}^{d} \sum_{j'=1}^{d} \sum_{m=1}^{c} \sum_{k=1}^{\text{len}(\mathbf{t}_j)} \bar{\alpha}_{j,j',k,m} \mathbf{1}_{[mcat(x_{i,j})=m \text{ and } x_{i,j'} \leq t_{j',k}]}.$$

Rearranging:

$$g_{\text{miss}}(\mathbf{x}_i; \bar{\beta}, \bar{\beta}^{\text{miss}}, \bar{\alpha}) = \bar{\beta}_0 + \sum_{j=1}^{d} f_j(x_j, \bar{\beta}_j)$$

$$+ \sum_{j=1}^{d} \sum_{m=1}^{c} \bar{\beta}_{j,m}^{\text{miss}} \mathbf{1}_{[mcat(x_{i,j})=m]}$$

$$+ \sum_{j=1}^{d} \sum_{j'=1}^{d} \sum_{m=1}^{c} \sum_{k=1}^{\text{len}(\mathbf{t}_j)} \bar{\alpha}_{j,j',k,m} \mathbf{1}_{[mcat(x_{i,j})=m \text{ and } x_{i,j'} \leq t_{j',k}]}.$$

We can pick the non-missing coefficients $\bar{\beta}$ for Equation (6) to match those from Equation (5), leaving $\bar{\beta}_0 + \sum_{j=1}^{d} f_j(x_j, \bar{\beta}_j) = \beta_0 + \sum_{j=1}^{d} f_j(x_j, \beta_j)$.

Now pick $\bar{\beta}_{j,m}^{\text{miss}} = 0$ except when $j = b$, and $\bar{\beta}_{b,1}^{\text{miss}} = \left( \beta_b d - \frac{\beta_b a \min_{i'}(s(\mathbf{x}_{i',-b}))}{\max_{i'}(s(\mathbf{x}_{i',-b})) - \min_{i'}(s(\mathbf{x}_{i',-b}))} \right)$. Then we have:

$$g_{\text{miss}}(\mathbf{x}_i; \bar{\beta}, \bar{\beta}^{\text{miss}}, \alpha) = \beta_0 + \sum_{j=1}^{d} f_j(x_j, \beta_j)$$

$$+ \mathbf{1}_{[mcat(x_{i,j})=m]} \left( \beta_b d - \frac{\beta_b a \min_{i'}(s(\mathbf{x}_{i',-b}))}{\max_{i'}(s(\mathbf{x}_{i',-b})) - \min_{i'}(s(\mathbf{x}_{i',-b}))} \right)$$

$$+ \sum_{j=1}^{d} \sum_{j'=1}^{d} \sum_{m=1}^{c} \sum_{k=1}^{\text{len}(\mathbf{t}_j)} \bar{\alpha}_{j,j',k,m} \mathbf{1}_{[mcat(x_{i,j})=m \text{ and } x_{i,j'} \leq t_{j',k}]}.$$

$$g_{\text{miss}}(\mathbf{x}_i; \bar{\beta}, \bar{\beta}^{\text{miss}}, \bar{\alpha}) = \beta_0 + \sum_{j=1}^{d} f_j(x_j, \beta_j)$$

$$+ \sum_{j=1}^{d} \sum_{j'=1}^{d} \sum_{m=1}^{c} \sum_{k=1}^{\text{len}(\mathbf{t}_j)} \bar{\alpha}_{j,j',k,m} \mathbf{1}_{[mcat(x_{i,j})=m \text{ and } x_{i,j'} \leq t_{j',k}]}.$$

$$+ \mathbf{1}_{[mcat(x_{i,j})=m]} \left( \beta_b d - \frac{\beta_b a \min_{i'}(s(\mathbf{x}_{i',-b}))}{\max_{i'}(s(\mathbf{x}_{i',-b})) - \min_{i'}(s(\mathbf{x}_{i',-b}))} \right)$$

Now, parameterize $\sum_{j'=1}^{d} \sum_{m=1}^{c} \sum_{k=1}^{\text{len}(\mathbf{t}_j)} \bar{\alpha}_{b,j',k,m} \mathbf{1}_{[mcat(x_{i,b})=m \text{ and } x_{i,j'} \leq t_{j',k}]}$ to match $s(x_{-b}) \frac{\beta_b a}{\max_{i'}(s(\mathbf{x}_{i',-b})) - \min_{i'}(s(\mathbf{x}_{i',-b}))}$ by, for each $j'$, setting

$$\bar{\alpha}_{b,j',k,1} = C_{j',k} \frac{\beta_b a}{\max_{i'}(s(\mathbf{x}_{i',-b})) - \min_{i'}(s(\mathbf{x}_{i',-b}))}$$

Set $\bar{\alpha}_{j,j',k,m} = 0$ for all $j \neq b$. Then we have:

$$g_{\text{miss}}(\mathbf{x}_i; \beta, \beta^{\text{miss}}, \alpha) = \beta_0 + \sum_{j=1}^{d} f_j(x_j, \beta_j)$$

$$+ \mathbf{1}_{[mcat(x_{i,j})=m]} \left( \beta_b d - \frac{\beta_b a \min_{i'}(s(\mathbf{x}_{i',-b}))}{\max_{i'}(s(\mathbf{x}_{i',-b})) - \min_{i'}(s(\mathbf{x}_{i',-b}))} \right)$$

$$+ \mathbf{1}_{[mcat(x_{i,b})=1]} s(x_{-b}) \frac{\beta_b a}{\max_{i'}(s(\mathbf{x}_{i',-b})) - \min_{i'}(s(\mathbf{x}_{i',-b}))}.$$

Now, as required, we have shown that Equations 6 and 5 are equivalent.

$\square$

# D Experimental Details

## D.1 Conditional Probability Table for Added MAR Missingness

In Section 4.1, we evaluated the predictive power of M-GAM when additional synthetic missingness was added to each dataset we studied. Let $X_1$ denote the variable missingness is being added to, $X_2$ another column from the dataset used to determine whether to add missingness, and $Y$ our target outcome. Let $M$ denote whether synthetic missingness is added to $X_1$, and let $Q_{X_2}(p)$ denote the $p$-th quantile of $X_2$. For a target missing rate $r$, Table D.1 shows the conditional probability of $X_1$ being missing given each value of $Y$ and $X_2$. Note that missingness does not depend on $X_1$, making this an MAR setting.

|  | $Y = 0$ | $Y = 1$ |
|---|---|---|
| $X_2 \geq Q_{X_2}(0.6)$ | $P(M = 1|Y, X_2) = 0$ | $P(M = 1|Y, X_2) = r$ |
| $X_2 < Q_{X_2}(0.6)$ | $P(M = 1|Y, X_2) = r$ | $P(M = 1|Y, X_2) = 0$ |

Table 1: The conditional probability table obeyed when adding synthetic missingness.

## D.2 Data Processing

We follow the structure from (Shadbahr et al., 2023) for running our imputation baselines and selecting train/test splits. To collect the data for MIMIC, we used the process in (Johnson et al., 2018), and additional steps by (Zhu et al., 2023) to convert the data to a single tabular dataset.

## D.3 Detailed Experimental Setup

For every GAM we fit (M-GAM, FastSparse, and non-L0 GAMs), we created an indicator for each of 8 quantiles (the 0.125 quantile, the 0.25 quantile, and so on). On FICO, we include missingness indicator and interaction terms as appropriate for four encodings of missingness: the three types in the original dataset (no information, no usable information, and no report available) and an added indicator which is true for any type of missingness. All other datasets contain only one type of missingness.

We fit all M-GAMs using FastSparse (Liu et al., 2022), and in all cases set the "max_support_size" variable to 100. This prevents the algorithm from exploring models with greater than 100 non-zero coefficients. For all experiments that did not report complete sparsity versus accuracy curves, we used 5-fold cross validation to select the value for the $\ell_0$ sparsity penalty. We searched over the following set of values for $\lambda$ for each GAM: $20, 10, 5, 2, 1, 0.5, 0.4, 0.2, 0.1, 0.05, 0.02, 0.01$, and $0.005$. We optimized for AUC on Breast Cancer and accuracy on all other datasets when using cross validation. We fit all non-sparse GAM's using SKLearn's implementation of logistic regression over binned data.

We evaluated the performance of a variety of classifiers on all datasets in Section 4.4. For each datasets, we used MICE to impute 10 distinct datasets, and fit a variety of predictive models (a logistic regression, an AdaBoost model (Freund & Schapire, 1997), a random forest (Breiman, 2001), a decision tree, a shallow neural network, and an XGBoost classifier Chen & Guestrin (2016)) on these datasets. For each baseline classifier, we also provide accuracy for a model fit with and without missingness indicators added via the SMIM procedure Van Ness et al. (2023). We used cross validation to select hyperparameters separately for each imputed training dataset, and ensembled the 10 models for each model class to produce a single predictive model. Cross validation was performed using 5 folds via GridSearchCV from SKLearn (Pedregosa et al., 2011), and the SKLearn implementation was used for each model class considered other than XGBoost. The hyperparameters we considered are:

- Logistic regression: {"C":[0.01, 0.1, 1, 10], "penalty": ("l2"),"max_iter": [10,000], "tol": [5e-2]}
- Random forest: {"n_estimators":[25, 50, 100, 200], "criterion": ["gini","entropy"]}
- AdaBoost: {"n_estimators":[10, 25, 50, 100, 200]}
- Decision tree: {"max_depth":[3, 5, 7, 9, None], "criterion":("gini", "entropy")}

- Neural Network: {"hidden_layer_sizes":[(50,), (100,), (200,), (50, 50), (50, 100), (100, 100), (100, 200), (200, 200)], "tol": [5e-2], "max_iter": [1000]}
- XGBoost: { 'n_estimators':[100, 500, 1000], 'gamma':[0, 0.1], 'lambda':[.5, 1, 2], 'alpha':[.5, 1, 2] }

We repeat this procedure for ten distinct train-test splits for each dataset considered.

## D.4 Computational Resources

All experiments were performed on an institutional computing cluster. All experiments that involved timing were conducted using one Tensor TXR231-1000R D126 Intel(R) Xeon(R) CPU E5-2640 v4 @ 2.40GHz (512GB RAM - 40 cores), except for MIWAE timing experiments, which use one NVIDIA Tesla P100 GPU. When runtime was not reported, experiments were run on whatever hardware was available on the cluster at that time.

# E Additional Experiments

## E.1 Additional Datasets

Throughout this section of the appendix, we will consider two new datasets in addition to the four datasets introduced in the main body of the paper (FICO, Breast Cancer, MIMIC, and Pharyngitis). We add the Chronic Kidney Disease Rubini et al. (2015) and Heart Disease Janosi et al. (1988) datasets from UCI, refered to simply as CKD and Heart Disease respectively. CKD consists of 400 samples with 24 features, and involves prediction of chronic kidney disease using medical features. Heart Disease concerns predicting heart disease using medical and demographic features, with 303 samples and 13 features. Note that the outcome in Heart Disease is an integer between 0 and 4; we binarize this label such that we classify no heart disease (0) versus any heart disease (1, 2, 3, 4). Both of tshese datasets contain only one missingness encoding.

## E.2 Evaluation of Alternative Imputation Methods

In the main body of this paper, we focused our evaluation of baseline classifiers on impute-then-predict using the MICE method for multiple imputation. However, a wide variety of multiple imputation methods are available. Using all six datasets, we evaluate the runtime and accuracy of four imputation methods: MICE (Van Buuren & Oudshoorn, 1999), MIWAE (Mattei & Frellsen, 2019), mean value imputation, and MissForest (Stekhoven & Bühlmann, 2012). We impute ten alternative datasets for each of ten distinct train-test splits for both FICO and Breast Cancer. For a given train-test split, we fit a model from each of the model classes described in D.3 for each imputed training dataset. We ensemble these ten models to produce a single predictive model per train-test split, and evaluate the test accuracy (FICO) or test AUC (Breast Cancer) of this ensembled model.

Figure 8 contains box-and-whiskers plots for the accuracy of each method considered across all six datasets. The imputation method does not generally have a substantial impact on the performance of the resulting impute-then-predict classifier. We also see that, across the two datasets not considered in the main body of the paper (CKD and Heart Disease) M-GAM continues to provide comparable accuracy to all baseline methods.

Figure 9 shows the time required to produce a predictive model under each method for each dataset. As in the main body of the paper, we show the sum of the time required to impute data and the time required to produce the most accurate model. We see that, despite resulting in models with comparable accuracy to those produced by MICE and Mean imputation, MissForest and MIWAE take longer to compute on the majority of datasets. On all six datasets datasets, M-GAM is at least an order of magnitude faster than all three multiple imputation methods, although mean imputation tends to be fastest.

## E.3 Scalability of M-GAM and Imputation Methods

This section studies how well each method scales in terms of the number of samples in the dataset. For each dataset, we take subsamples of increasing size (25%, 50%, 75%, and 100% of samples in each dataset) and run each impute-then-predict predict procedure, as well as M-GAM over 10 distinct train-test splits. Figure 10 reports the total time taken to produce a model for each imputation method and M-GAM on each dataset/subsample combination. We find that M-GAM scales no worse than any of the imputation alternatives in terms of runtime.

## E.4 Evaluation of Different Thresholds

Throughout the main body of this work, we reported results using 8 evenly spaced quantiles to threshold our input variables for both M-GAM and FastSparse GAMs fit on imputed data. In this section, we evaluate the sparsity versus accuracy curve for M-GAM under different binning strategies. In particular, we evaluate the performance of M-GAM on FICO with 4, 8, 16, and 32 evenly spaced quantiles. Figure 11 shows the results of this analysis. As the number of quantiles increased, M-GAM remained sparse despite the exploding number of interaction terms, and in fact for 32 quantiles the interaction terms lead to an especially sparse and accurate model. Beyond these observations, we found that reasonable changes to the number of thresholds we consider did not significantly impact performance.

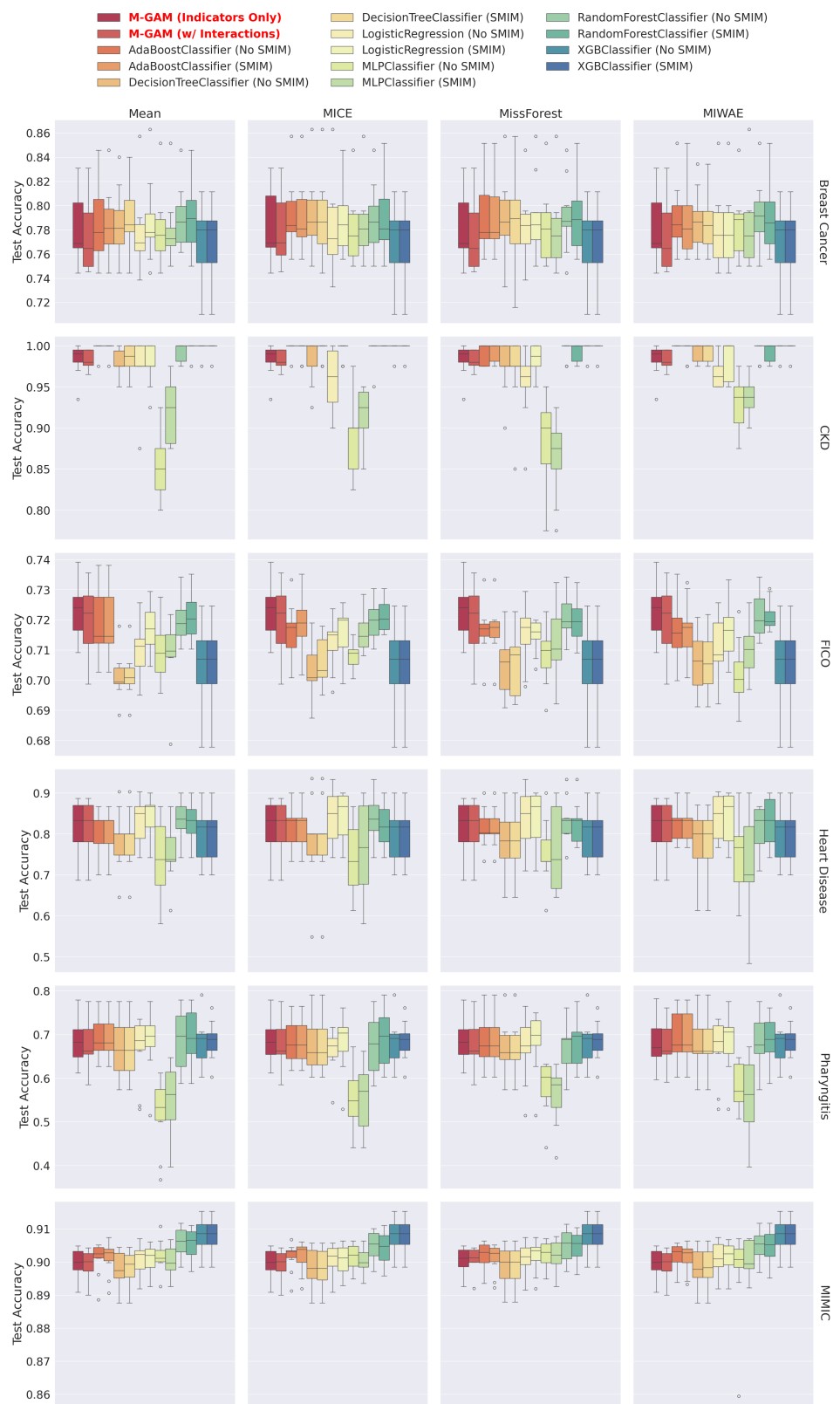

Figure 8: Test accuracy of M-GAM compared against a variety of baselines for four imputation methods and six datasets. Each column corresponds to a different imputation method, and each row to a different dataset.

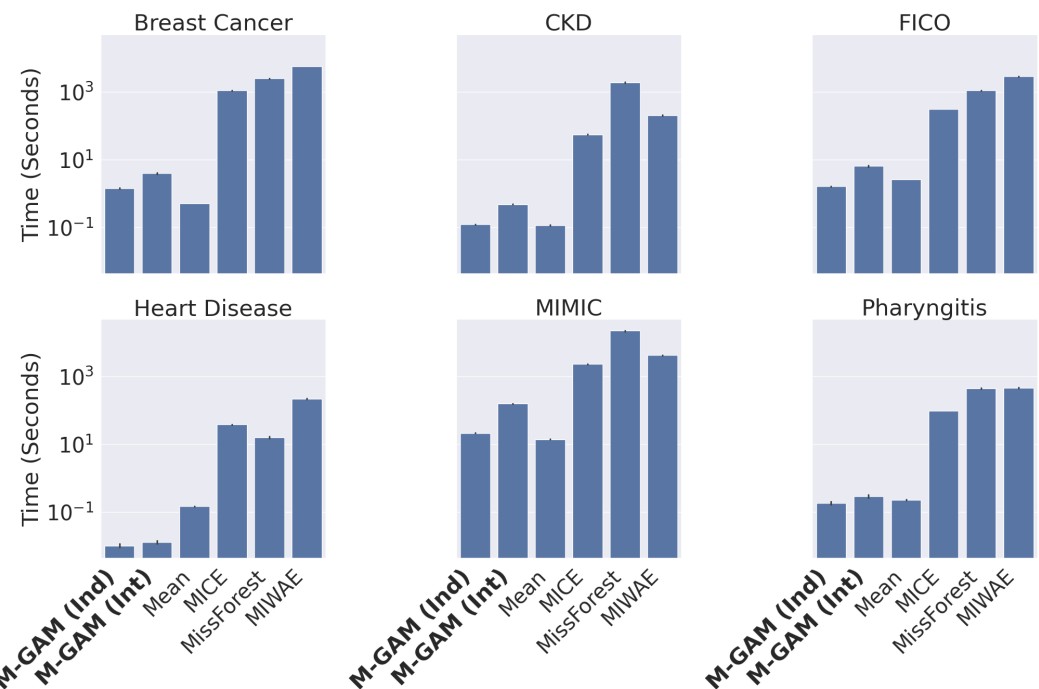

Figure 9: Runtime of different methods on Breast Cancer, CKD, FICO, Heart Disease, MIMIC, and Pharyngitis. For each imputation method, we report the total time required to impute missing data and fit the best performing impute-then-predict classifier for that dataset and imputation method. M-GAM (Ind) is an M-GAM with indicators and M-GAM (Int) is an M-GAM with interaction terms. Error bars report standard error of total runtime over 10 train-test splits.

### E.5  Evaluation of MICE with Different Numbers of Imputations

Since MICE is a *multiple* imputation method, we needed to choose how many datasets we allow MICE to impute for each of our experiments. In this section, we evaluate the runtime versus test accuracy for models built on various numbers of imputed datasets for FICO. We evaluated each non-GAM baseling model considered in the main paper when ensembled over 1, 5, 10, 20, and 30 MICE imputed datasets.

Figure 12 shows the accuracy versus runtime for each number of imputations. In Figure 12, we see that there is a slight improvement in the accuracy of our classifiers when increasing from 1 to 5 imputations, but no significant performance gain for any larger numbers of imputed datasets. As such, we opted to use the moderately fast and performant choice of 10 imputations.

### E.6  Extension of Sparsity/Accuracy Results to Further Datasets

We focus on the FICO and Breast Cancer datasets for much of the main paper, alongside Pharyngitis and MIMIC. In Figure 13 we show the superset of our sparsity-accuracy results that includes the two UCI repository datasets, Heart Disease Janosi et al. (1988) and CKD Rubini et al. (2015). In Figure 14 we show the superset of our results for the data with added MAR missingness, for all 6 datasets.

### E.7  Evaluation of Alternative Distinct Missingness Encodings

As we discuss in the main text, allowing our model to encode different reasons for missingness allows the ability to handle multiple reasons for missingness, improving the model's power. However, using only a single, overall reason for missingness could potentially allow for handling a larger set of missing data cases with fewer coefficients. These two encodings are not mutually exclusive; we could augment in both ways, as described in Equation E.1. For this reason, we investigate sparsity and test accuracy on the FICO dataset across a range of choices for whether to use distinct encodings,

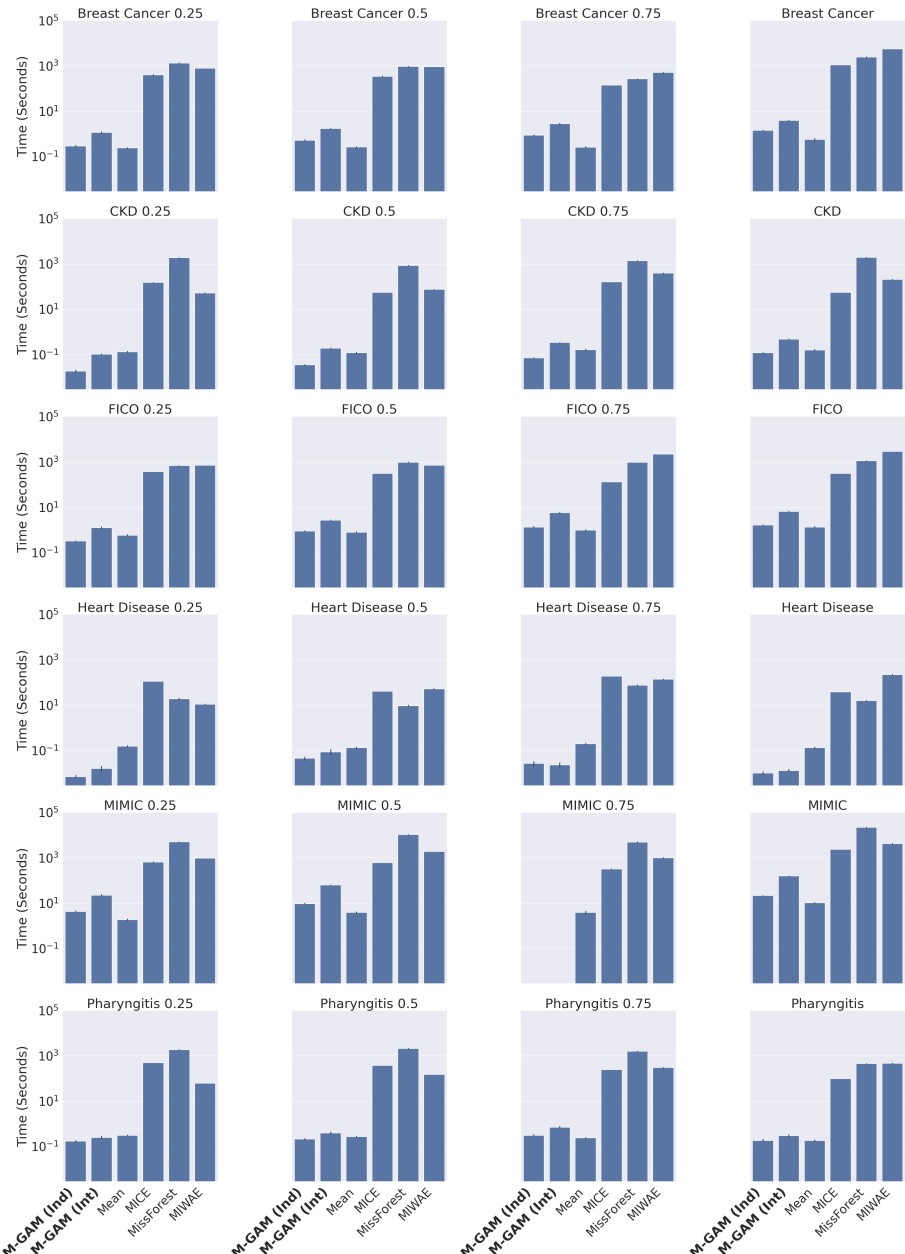

Figure 10: Runtime of different methods over 25%, 50%, 75%, and all of Breast Cancer, CKD, FICO, Heart Disease, MIMIC, and Pharyngitis. For each imputation method, we report the total time required to impute missing data and fit the best performing impute-then-predict classifier for that dataset and imputation method. M-GAM (Ind) is an M-GAM with indicators and M-GAM (Int) is an M-GAM with interaction terms. Error bars report standard error of total runtime over 10 train-test splits.

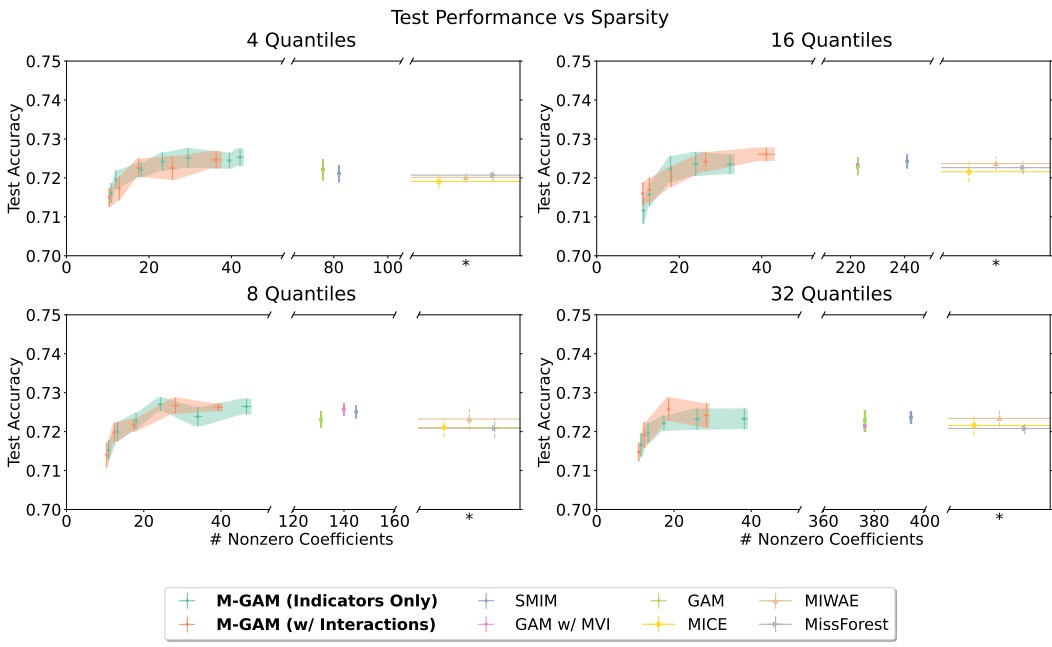

Figure 11: The sparsity versus accuracy plot for four distinct binning strategies for M-GAM on the FICO dataset.

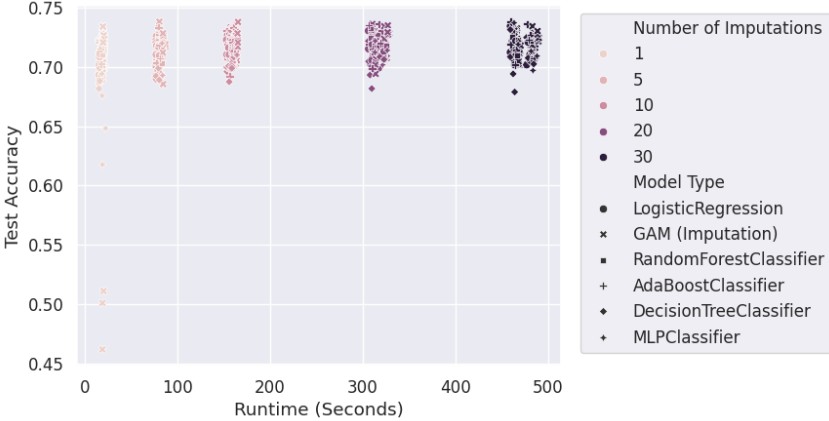

Figure 12: Test accuracy versus runtime for models built on different numbers of MICE imputed datasets. Each color represents a different number of imputed datasets, and each shape represents a different ensembled model fit on these datasets.

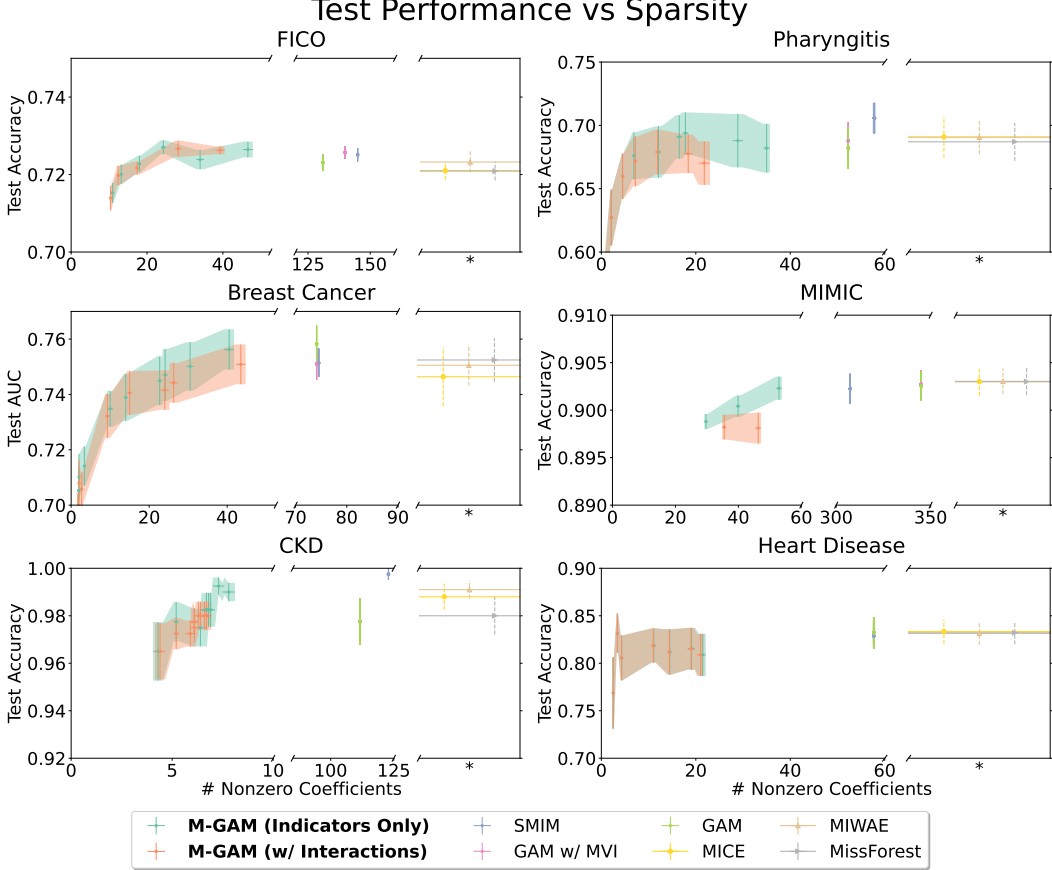

Figure 13: Test accuracy vs sparsity for M-GAMs relative to competitor GAMs on 6 datasets.

overall encodings, or some combination of the two. We explore a variety of combinations in Figure 15, allowing nonzero values for different subsets of $\alpha$ (specific indicators), $\alpha^{\text{overall}}$ (overall indicators), $\beta^{\text{miss}}$ (specific interactions), and $\beta^{\text{overall\_miss}}$ (overall interactions). We find no dramatic differences across these choices.

**Definition E.1.** Given parameters $\alpha$, $\alpha^{\text{overall}}$, $\beta^{\text{miss}}$, $\beta^{\text{overall\_miss}}$, and $\beta$, an M-GAM is defined as

$$g_{\text{miss}}(\mathbf{x}_i; \beta, \beta^{\text{miss}}, \beta^{\text{overall\_miss}}, \alpha, \alpha^{\text{overall}}) = \beta_0 + \sum_{j=1}^{d} h_j(x_{i,j}; \beta_j, \beta_j^{\text{miss}}, \beta_j^{\text{overall\_miss}}) \quad (7)$$

$$+ \sum_{j=1}^{d} \sum_{j'=1}^{d} h_{j,j'}(x_{i,j}, x_{i,j'}; \alpha_{j,j'}, \alpha_{j,j'}^{\text{overall}}),$$

where

$$h_{j,j'}(x_{i,j}, x_{i,j'}; \alpha_{j,j'}, \alpha_{j,j'}^{\text{overall}}) = \sum_{m=1}^{c} \sum_{k=1}^{\text{len}(\mathbf{t}_j)} \alpha_{j,j',k,m} \mathbf{1}_{[mcat(x_{i,j})=m \text{ and } x_{i,j'} \leq t_{j',k}]}$$

$$+ \sum_{k=1}^{\text{len}(\mathbf{t}_j)} \alpha_{j,j',k}^{\text{overall}} \mathbf{1}_{[mcat(x_{i,j}) \neq 0 \text{ and } x_{i,j'} \leq t_{j',k}]}$$

and

$$h_j(x_{i,j}; \beta_j, \beta_j^{\text{miss}}, \beta^{\text{overall\_miss}}) = f_j(x_{i,j}; \beta_j) + \sum_{m=1}^{c} \beta_{j,m}^{\text{miss}} \mathbf{1}_{[mcat(x_{i,j})=m]}$$

$$+ \beta_j^{\text{overall\_miss}} \mathbf{1}_{[mcat(x_{i,j}) \neq 0]}$$

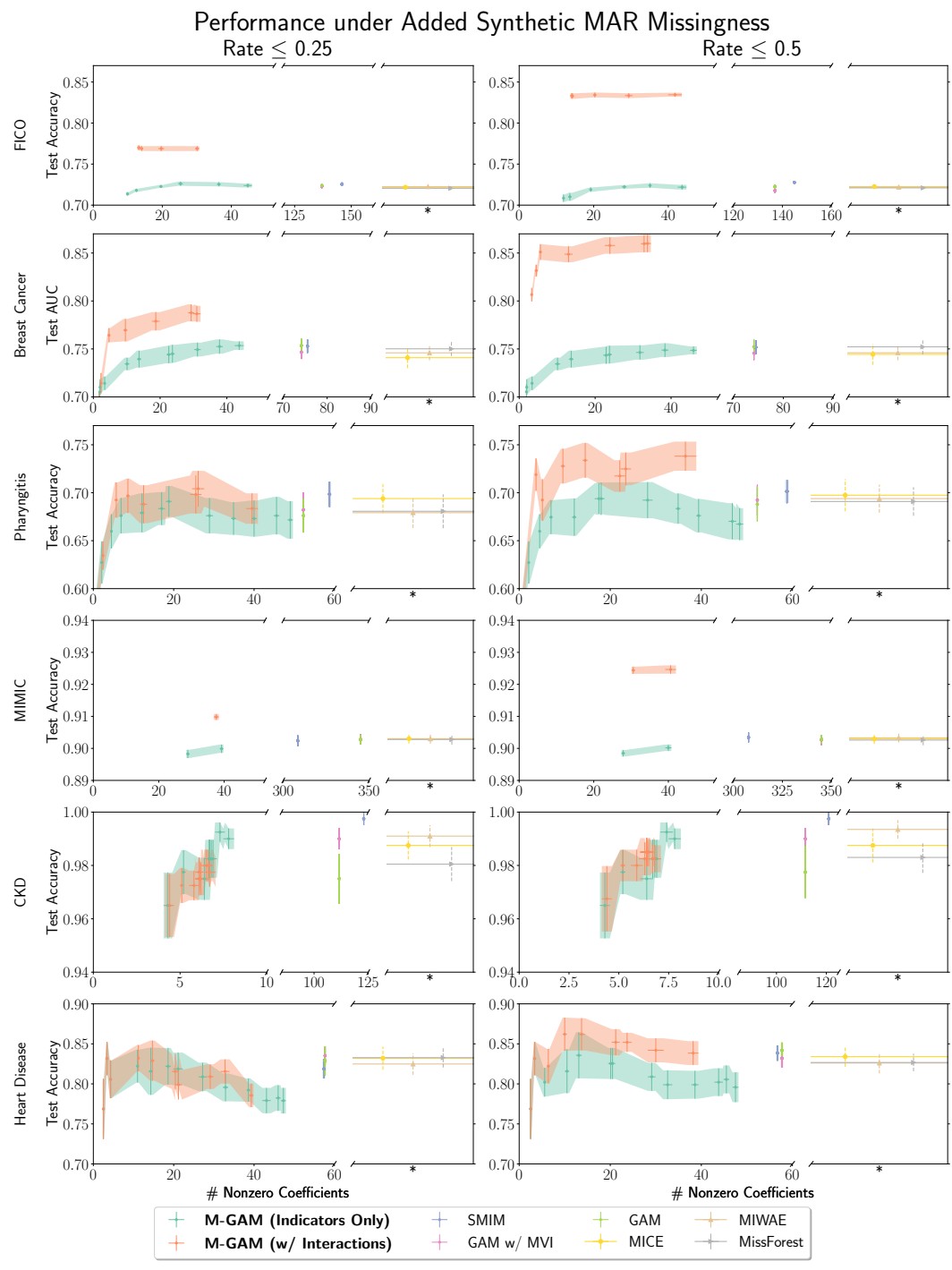

Figure 14: Test accuracy vs sparsity for M-GAMs relative to competitor GAMs on 6 datasets with added missingness.

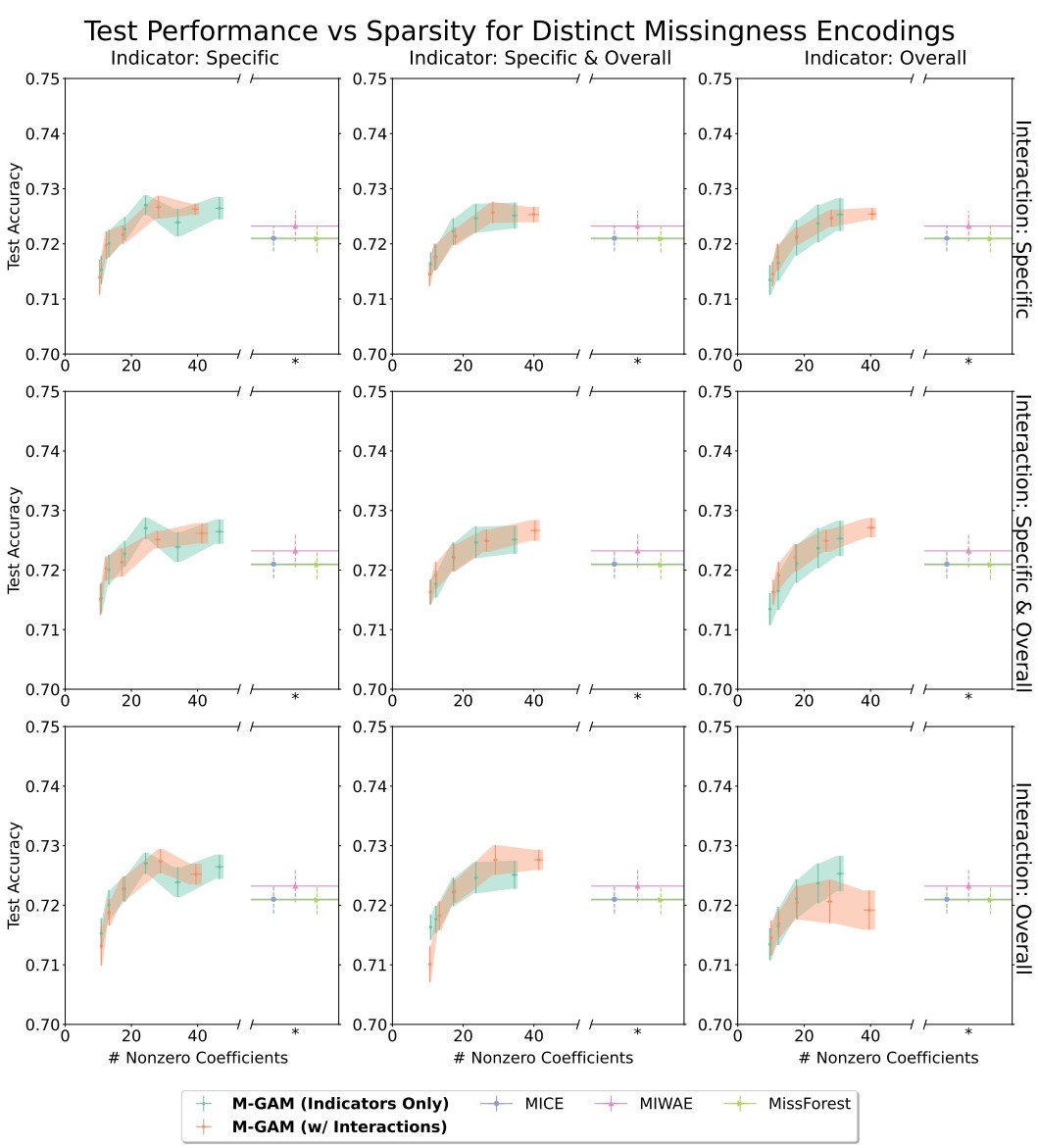

Figure 15: Results on the FICO dataset using different choices of missingness augmentation for indicators and interactions. "Specific" refers to the distinct missingness used throughout the text. "Overall" refers to augmenting our matrix while treating missing data as all having a single missingness reason.

# F   Additional M-GAM Visualizations

In this section, we visualize three additional M-GAM, one with and one without missingness interactions on FICO (Figures 17 and 18, respectively), and one without missingness interactions on Breast Cancer (Figure 19. These figures are best viewed digitally.

Note that several shape functions in Figure 19 are simply flat lines; this is because several variables in Breast Cancer (e.g., "M Stage" and "Overall Patient Receptor Status Triple Negative") are binary.

# G   License Information for Used Assets

In this section, we provide license information for every external asset used in this paper. The Breast Cancer, CKD, Heart Disease, and Pharyngitis datasets are available under creative commons. FICO is used under its own license, the details of which can be found here: `https://community.fico.com/s/explainable-machine-learning-challenge?tabset-158d9=2`. MIMIC-III is available under the MIT license. We use code from Liu et al. (2022) under the MIT license, and the code from Shadbahr et al. (2023) under BSD 3-Clause.

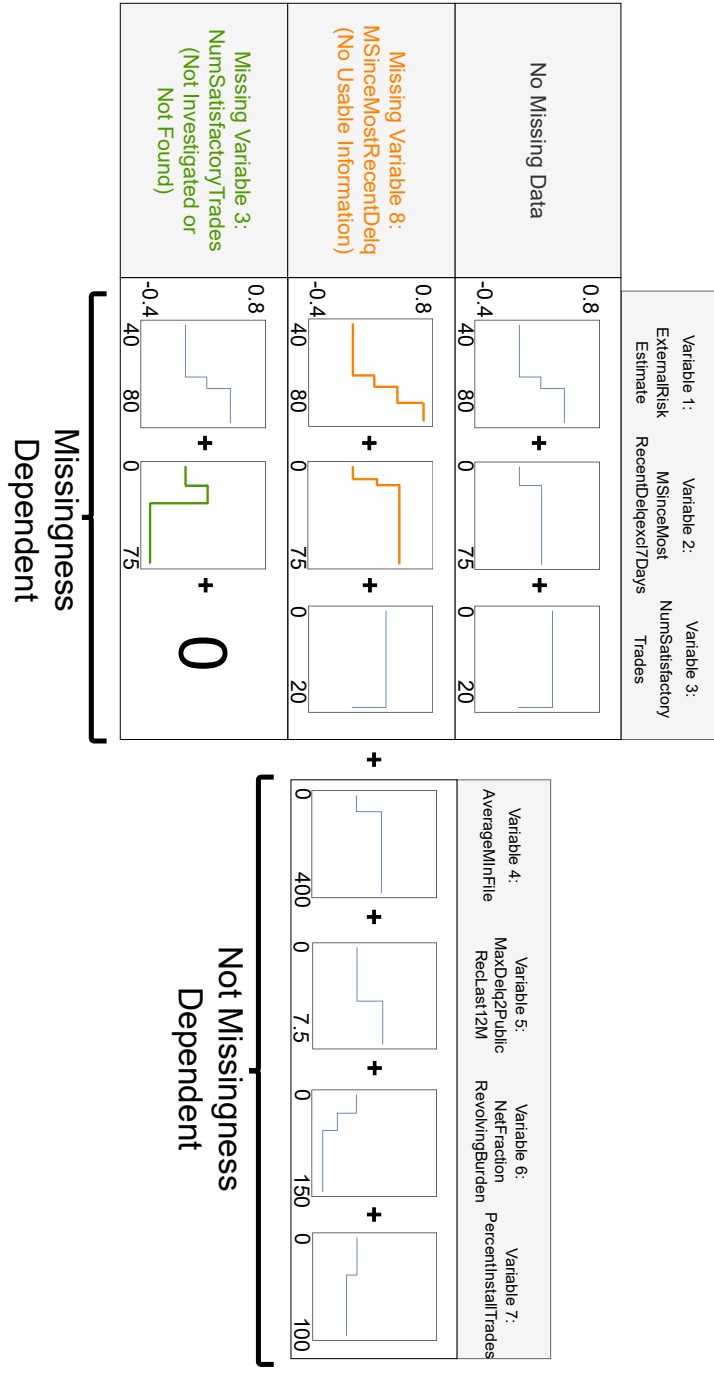

Figure 16: An expanded version of Figure 2 with variable names included. "MSinceMostRecentDelq" is the number of months since the individual's last delinquent payment and "MSinceMostRecentIn-qexcl7day" is the number of months since the individual's last inquiry, excluding those within the last week; all features are described in the FICO challenge (FICO et al., 2018) data documentation.

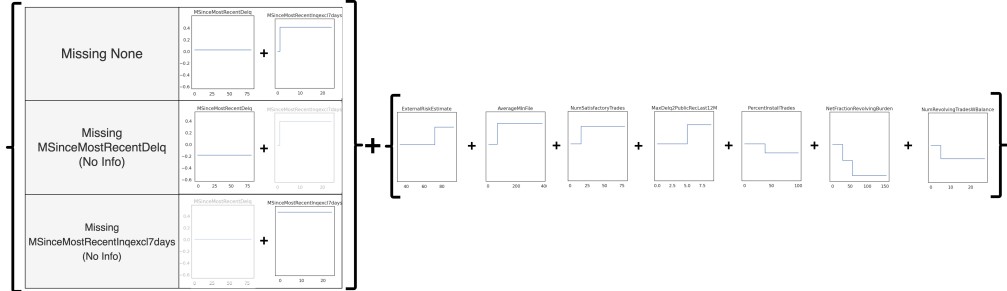

Figure 17: A visualization of a M-GAM without interaction terms on FICO. The shape functions on the left are selected based on which variables are missing, with the relevant missing variable noted to the left. The shape functions on the right are used in all cases.

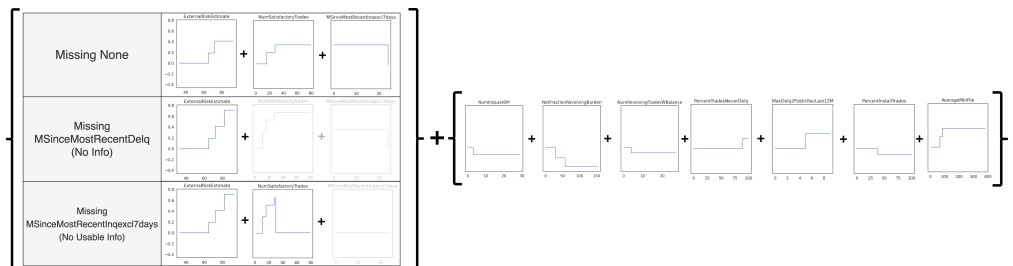

Figure 18: An additional visualization of a M-GAM with interaction terms on FICO. The shape functions within the left set of brackets are selected based on which variables are missing, with the relevant missing variable noted to the left. The shape functions in the right set of brackets are applied in all cases.

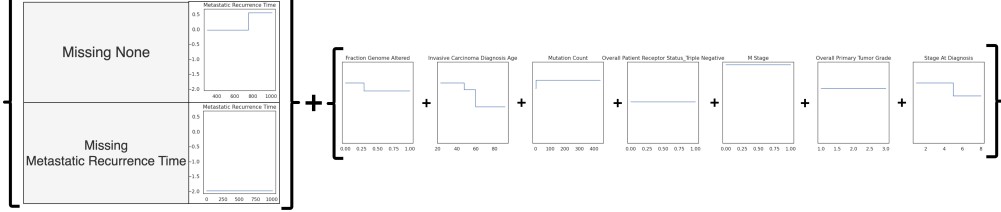

Figure 19: A visualization of a M-GAM without interaction terms on Breast Cancer. The shape functions within the left set of brackets are selected based on which variables are missing, with the relevant missing variable noted to the left. The shape functions in the right set of brackets are applied in all cases.

