# OpenReview forum: "Interpretable Generalized Additive Models for Datasets with Missing Values"
_NeurIPS.cc/2024/Conference — NeurIPS 2024 poster_

### Official Review · Reviewer_W4vf · 2024-07-07

**Soundness:** 2
**Presentation:** 2
**Contribution:** 2
**Rating:** 5
**Confidence:** 3

**Summary:**

The paper discusses challenges posed by missing data in important datasets for machine learning models. Existing methods like imputation or using indicator variables for missingness can compromise model interpretability or introduce complexity and reduced sparsity. The authors propose M-GAM, a sparse, generalized additive modeling approach. M-GAM addresses these issues by incorporating missingness indicators and their interactions, while maintaining sparsity through l0 regularization. They demonstrate that M-GAM achieves comparable or better accuracy than existing methods while significantly improving sparsity compared to imputation or straightforward inclusion of indicator variables.

**Strengths:**

- The paper is well-written

- The proposed method is novel

- Extensive experiments are conducted to robustly support the claims.

**Weaknesses:**

- The proposed method is constrained by its reliance on l_0 regularization.

**Questions:**

N/A

---

> ### Author Rebuttal · Authors · 2024-08-06
>
> Thank you for your review! We appreciate your recognition that this work is well written, novel, and contains extensive experiments, and we are open to discussing any additional concerns you might have.
>
> > The proposed method is constrained by its reliance on l_0 regularization.
>
> While $\ell_0$ regularization has historically been difficult to optimize, a key strength of our paper is that it leverages the fast optimization framework from Liu et al. 2022, which introduces several computational tricks that make it quite manageable. This lets us gain the substantial benefits of $\ell_0$ regularization (extreme sparsity in a setting that is prone to producing dense models) without suffering substantial runtime costs.

---

> > ### Comment · Reviewer_W4vf · 2024-08-13
> >
> > Thank you for your response. I've decided to keep my original score.

---

> > > ### Author Response · Authors · 2024-08-13
> > >
> > > Thank you again for your feedback!

---

### Official Review · Reviewer_63xg · 2024-07-09

**Soundness:** 3
**Presentation:** 3
**Contribution:** 2
**Rating:** 6
**Confidence:** 4

**Summary:**

The paper presents M-GAM, a novel generalized additive model that that incorporates missingness indicators while maintaining a sparse model via l_0 regularization. Results shows that on augmented datasets with missing at randomness M-GAM provides better performance; on real-world (not augmented) datasets, M-GAM achieves similar performance but is faster than impute-then-predict.

**Strengths:**

- Novel approach for dealing with missing values in GAMs while keeping the model sparse.
- Interesting theoretical results to support the work including
- The paper is generally clear and written well.

**Weaknesses:**

- My understanding is that the setting considered in this paper seems to be limited to GAMs with main effects and not higher-order GAMs (in particular, GAM with pairwise interaction effects, GA2M [Lou et al., KDD 2013]).
- There has been significant work on handling with missing values in decision trees (that are also a widely used interpretable model). There is very little both in terms of discussion (e.g., are the approaches there applicable to GAMs, does the proposed approach share some similarities to existing approaches in decision trees) as well as in terms of experimental results (how do decision trees with missing values compare to GAM with missing values, at the moment only SMIM with decision tree is considered and not approaches that are inherent to decision trees).
- The experimental results on real datasets do not demonstrate strong improvement in performance (but there is improvement in other factors, e.g., runtime compared to impute-then-predict).
- The analysis on other models, including decision trees (with and without SMIM) does not include runtimes so it is not clear if the new model is also significantly faster from, say, decision trees or logistic regression or random forests with SMIM.

**Questions:**

I would appreciate the authors response to the weaknesses listed above. In addition I was wondering if the authors have considered incorporating this mechanism into one of the recent neural GAM approaches, e.g., Neural Additive Models [] which will allow potentially for better scalability?

**Limitations:**

Limitations are adequately addressed.

---

> ### Author Rebuttal · Authors · 2024-08-06
>
> We appreciate your review, and your recognition of this work’s novelty, theoretical foundations, and clear writing. We have responded to each of your criticisms below, and look forward to any continued discussion.
>
> > My understanding is that the setting considered in this paper seems to be limited to GAMs with main effects and not higher-order GAMs (in particular, GAM with pairwise interaction effects, GA2M [Lou et al., KDD 2013]).
>
> This is generally correct, although M-GAMs could naturally be extended to include interaction effects of an arbitrary order – both between missingness and between observed values, and between different combinations of observed values. However, we note that, even in Lou et al. 2013, higher order GAMs showed substantially improved classification performance relative to main effect GAMs only on digit recognition tasks (“Letter”, “Gisette”) and not on datasets that would be treated as tabular today. We did experiment with additional interaction terms, along the lines of GA2M, but did not find any scenarios where this improved performance. As such, we did not feel comfortable including it in the paper.
>
> >There has been significant work on handling with missing values in decision trees (that are also a widely used interpretable model). There is very little both in terms of discussion (e.g., are the approaches there applicable to GAMs, does the proposed approach share some similarities to existing approaches in decision trees) as well as in terms of experimental results (how do decision trees with missing values compare to GAM with missing values, at the moment only SMIM with decision tree is considered and not approaches that are inherent to decision trees).
>
> This is a fair point. We will add discussion of this work in the related work section, and have added comparisons to some decision tree based methods that explicitly handle missingness as implemented in SKLearn, as shown in the shared response. In general, because these papers introduce changes that are specific to decision trees, such as default traversal paths when data is missing, they are not directly applicable to M-GAM. We found that decision trees with this explicit handling improved runtime and offered interpretability, but generally had poorer performance than M-GAM. Random forests offered comparable performance and runtime, at the cost of interpretability.
>
> In summary, we’re happy to address this work in related work, but it does not decrease the value of our own approach, which performs well and is separate from the decision-tree-inherent methods you’ve mentioned.
>
> > The experimental results on real datasets do not demonstrate strong improvement in performance (but there is improvement in other factors, e.g., runtime compared to impute-then-predict).
>
> While this is true, the fact that M-GAM has comparable performance to other methods on real data means that our runtime and interpretability gains come with no substantial costs. M-GAM quickly provides a sparse, transparent model that is as accurate as complex impute-then-predict baselines, which we see as a strength.
>
> > The analysis on other models, including decision trees (with and without SMIM) does not include runtimes so it is not clear if the new model is also significantly faster from, say, decision trees or logistic regression or random forests with SMIM.
>
> We have added some such comparisons in the global response. Please note that the SMIM framework consists of both imputing missing data and providing missingness indicators, meaning that it is no faster than the given runtimes with imputation. Additionally, these methods that use imputation introduce complexity, sacrificing interpretability in a way that M-GAM avoids.
>
> > I would appreciate the authors response to the weaknesses listed above. In addition I was wondering if the authors have considered incorporating this mechanism into one of the recent neural GAM approaches, e.g., Neural Additive Models [] which will allow potentially for better scalability?
>
> We hadn’t previously, but it's an interesting idea. However, based on the results presented in [1], it seems that NAM’s will not actually scale better than M-GAMs. Our current $\ell_0$ regularized method is already quite fast.  If we were to discard our approach in favor of a Neural GAM with some other sparsity approach, we don’t anticipate benefits to scalability because such an approach  would effectively need to fit neural nets for up to (# features)$^2$ shape functions to cover missingness interactions, and a sparsity-regularized neural GAM can take quite a while even for a simple dataset. For example, table 5 of [1] shows a neural GAM taking ~30 seconds on a 6172 sample, 13-feature dataset, COMPAS, which is slower than our method on the much more computationally difficult FICO dataset (10,459 samples, 23 features), even when we augment the FICO dataset with missingness interactions.
>
> [1]  Shiyun Xu, Zhiqi Bu, Pratik Chaudhari, Ian J. Barnett. Sparse Neural Additive Model: Interpretable Deep Learning with Feature Selection via Group Sparsity. ECML PKDD 2023

---

> > ### Comment · Reviewer_63xg · 2024-08-13
> >
> > Thank you for your response. I will increase my score by one point.

---

> > > ### Author Response · Authors · 2024-08-13
> > >
> > > Thank you again for your feedback, and for increasing your score!

---

> > > > ### Author Response · Authors · 2024-08-13
> > > >
> > > > Once again, thank you for your review, and for raising your score. We've noticed that it seems like the rating update hasn't gone through on OpenReview. Would you mind updating this score? Thank you!

---

> > > > > ### Comment · Reviewer_63xg · 2024-08-13
> > > > >
> > > > > Updated now.

---

### Official Review · Reviewer_fBin · 2024-07-09

**Soundness:** 3
**Presentation:** 3
**Contribution:** 2
**Rating:** 6
**Confidence:** 3

**Summary:**

The paper introduces M-GAM, a novel extension of Generalized Additive Models (GAMs) designed to maintain interpretability while handling datasets with missing features. By incorporating missingness indicators and their interaction terms through ℓ0 regularization, M-GAM balances accuracy and sparsity. The model provides comparable or superior performance to traditional imputation methods while avoiding the complexity and overfitting issues associated with the naive inclusion of missingness indicators.

**Strengths:**

- Very clear explanation in section 3.
- Sparsity and interpretability are highly relevant for practitioners.

**Weaknesses:**

- The title should convey that the study is limited to GAMs
Runtime comparison in 4.3 is a bit vacuous because it depends on the exact implementation, etc. Maybe looking at computational complexity would be more interesting.
- Empirical evaluation is somewhat limited

**Questions:**

- l.27, it seems strange to contrast the choice of (not) using missingness with the choice of using l0 regularization. They seems to be rather orthogonal choices.
- Why l0 and not some other regularization?
- It seems questionable to encode reasons for missingness with natural numbers, as these imply an ordering.
- It looks like all other methods have better mean accuracy than M-GAM in Figure 6, Breast Cancer dataset. Is this correct? If so, the text should discuss this rather than glossing over it.

**Limitations:**

- The authors state interpretability as one of M-GAM's main advantages. I would like to see this made explicit, e.g., with an example, especially in comparison to existing methods.
- Empirical performance is not great.

---

> ### Author Rebuttal · Authors · 2024-08-06
>
> Thank you for the thoughtful review – we appreciate your recognition that sparsity and interpretability are particularly important for practitioners. We hope we have addressed each of your concerns below, and look forward to any ongoing discussion.
>
> > The title should convey that the study is limited to GAMs
>
> This is a fair point – we plan to update the title to something like  “Interpretable Generalized Additive Models for Datasets with Missing Values”. GAMs are currently among the most popular forms of interpretable ML models [1].
>
> [1] https://github.com/interpretml/interpret
>
> > Runtime comparison in 4.3 is a bit vacuous because it depends on the exact implementation, etc. Maybe looking at computational complexity would be more interesting.
>
> While computational complexity would provide an interesting additional kind of comparison, we believe practical runtime is the more relevant metric here because it is more reflective of what users will experience. This is particularly true for methods like FastSparse, where computational tricks substantially improve the algorithm’s practical runtime without necessarily affecting its big O runtime.
> While runtime can vary with implementation, we deliberately used implementations that are well established in prior work whenever possible. As such, the results in 4.3 should be fairly reflective of what an actual user would experience, as they would likely use these same implementations.
>
> > Empirical evaluation is somewhat limited
>
> We are unsure in what way this is meant, but note that appendix E contains a variety of additional empirical results, including evaluation over six datasets using a wide variety of baseline methods. Together, sections D and E of the appendix are longer than the entire main paper and consist of empirical evaluation.
>
> > l.27, it seems strange to contrast the choice of (not) using missingness with the choice of using l0 regularization. They seems to be rather orthogonal choices.
>
> Yes, they are orthogonal choices. We think this is just ambiguous phrasing – we meant to contrast handling missingness alongside $\ell_0$ regularization with handling missingness without $\ell_0$ regularization. We will update the wording to make this clearer.
>
> > Why l0 and not some other regularization?
>
> $\ell_0$ regularization is beneficial because it produces very sparse models, and this paper is focused on interpretability. In the context of M-GAMs, this means fewer variables will contribute to model predictions and therefore fewer shape functions will need to be presented. This makes it easier for users to interpret model predictions.
>
> > It seems questionable to encode reasons for missingness with natural numbers, as these imply an ordering.
>
> Good point, we’ll change it to letters (or another notation you suggest that implies no ordering). This does not actually affect the implementation since M-GAM is only exposed to the binary indicator variable for each missingness reason. If this is referring to the FICO description from the dataset description, it was not our choice to use the stated missingness encodings; those are from the creators of the dataset.
>
> > It looks like all other methods have better mean accuracy than M-GAM in Figure 6, Breast Cancer dataset. Is this correct? If so, the text should discuss this rather than glossing over it.
>
> This is true, but reflects an oversight in creating Figure 6. Breast Cancer and MIMIC are both heavily imbalanced, and as such performance is better measured via AUC, which is done in other plots. Accuracy is not reflective of quality for imbalanced datasets. Under this metric, it is not true that the median AUC for M-GAM is worse than all other methods, as shown in the combined response.
> The class balance for each of the four datasets is as follows
>
> Breast Cancer: 78.2% negative
>
> MIMIC: 89.6% negative
>
> Pharyngitis: 54.9% negative
>
> FICO: 47.8% negative
>
> > The authors state interpretability as one of M-GAM's main advantages. I would like to see this made explicit, e.g., with an example, especially in comparison to existing methods.
>
> We intended for Figures 1 and 2 to serve as such examples, but believe that additional examples on the datasets we study might help prove this point. We will prepare such examples, and add them to the appendix.

---

> > ### Comment · Reviewer_fBin · 2024-08-12
> > **Ack**
> >
> > Thank you for the response. With the additional explanation and background (provided also in responses to other reviewers), I would like to up my overall rating to 6. I hope this paper will be accepted.

---

> > > ### Author Response · Authors · 2024-08-12
> > >
> > > Thank you again for your constructive review, and for increasing your score!

---

> > > > ### Author Response · Authors · 2024-08-13
> > > >
> > > > Once again, thank you for your review, and for raising your score. We've noticed that it seems like the rating update hasn't gone through on OpenReview. Would you mind updating this score? Thank you!

---

### Official Review · Reviewer_fLWT · 2024-07-11

**Soundness:** 3
**Presentation:** 2
**Contribution:** 3
**Rating:** 5
**Confidence:** 3

**Summary:**

This paper introduces M-GAM, which incorporates concept of missingness into Generalized Additive Model(GAM). Since GAM represents arbitrary function with sum of univariate functions which take each input feature as input, M-GAM maintains sparsity and interpretability for inference with missing data.

**Strengths:**

- Adapting GAM as tool for modeling under missingness is new and useful. Common approach for integrating missingness information is concatenating missingness indicator to input feature, but this makes exhaustive expansion of input dimension which can potentially harm the inference in many aspects. So I think this approach is smart and efficient
- it is faster than other baselines with reasonable performance
- abundant experiments that support the authors claim and detailed description of experimental setup

**Weaknesses:**

- I think the main weakness of M-GAM is its performance which just barely match other baselines' performance. Since it is trained end-to-end manner, we can expect more performance gain than impute-then regress methods since it directly uses supervision during training. Since informative missingness is rare in real world dataset, predictive performance of M-GAM  on real world dataset can harm the applicability.

**Questions:**

- why M-GAM especially good on MAR setup?

**Limitations:**

Yes

---

> ### Author Rebuttal · Authors · 2024-08-06
>
> Thank you for your review. We appreciate your recognition of the novelty of this work and its strong experimental backing. We look forward to discussing any ongoing questions or concerns you may have.
>
> > I think the main weakness of M-GAM is its performance which just barely match other baselines' performance. Since it is trained end-to-end manner, we can expect more performance gain than impute-then regress methods since it directly uses supervision during training. Since informative missingness is rare in real world dataset, predictive performance of M-GAM on real world dataset can harm the applicability.
>
> The goal of this work is to improve interpretability without harming performance, and we accomplished that. M-GAM is consistently among the most performant models on each dataset. Sometimes, it even improves both performance and interpretability. By producing an interpretable model, we allow practitioners to easily identify confounded reasoning and use the model more responsibly. This ability to troubleshoot helps improve overall accuracy of the system during and after deployment, not just performance on a static dataset.
>
> > why M-GAM especially good on MAR setup?
>
> M-GAM is particularly effective in the MAR setup used in the semi-synthetic example because the value is MAR with respect to the label. That is, in that semi-synthetic case, there is some signal about the outcome embedded in whether or not a feature is missing.
>
> This can happen in a variety of settings; for example, imagine conducting a survey to help predict whether individuals care about privacy or not. If we ask for a home address and an individual doesn’t answer the question, resulting in missing data, that might be a strong indicator that they do care about privacy.

---

> > ### Comment · Reviewer_fLWT · 2024-08-09
> >
> > Thanks for answering my questions. Your answers are really helpful for my understanding of this paper.

---

> > > ### Author Response · Authors · 2024-08-09
> > >
> > > Thanks again for your feedback, we're glad this was helpful! We're happy to discuss any further issues you might have, and we hope you will consider increasing your score if we have addressed them all :)

---

### Official Review · Reviewer_i65A · 2024-07-13

**Soundness:** 3
**Presentation:** 3
**Contribution:** 2
**Rating:** 5
**Confidence:** 3

**Summary:**

The paper "Interpretable Machine Learning for Datasets with Missing Values" proposes generalized additive models incorporating missingness indicators and their interaction terms with sparseness ensured by l0 regularization. Therefore, the authors combine GAMs with the Missing indicator method and l0 regularization. The method is compared on real-world data sets with missing values and added synthetic missingness and against a variety of techniques used on imputed data with and without selective addition of missingness indicators.

**Strengths:**

The manuscript is well-written and structured.
Utilizing models of missingness into the ML process is a nice idea.

**Weaknesses:**

I would have run 2 distinct experiments at least. One with synthetic missingness where the ground truth is known and the missigness can be increased. Additionally I would have liked to see also a mix of MCAR and MAR to demonstrate the results when the assumed mechanism is incorrectly chosen.
The authors emphasize the interpretability enabled by M-GAM and figures are shown. It would have been nice to discuss the interpretation in the text.

**Questions:**

Proof only works since the unobserved noise for k1 and k2 is chosen as it is. "Let k_2<k_1<0.5"
To me this looks like circular reasoning. If I assume the noise is lower for knowing the missingness relationship with the predictor Y, than my uncertainty with the oracle -> of course the one with lower noise with reach higher probability. If the assumption is switched or equal signs added the proposition will fall apart. It seems the proposition needs to be far more limited than it is right now.
In 108 should it than not say "Corollary 3.2 states that perfectly imputing missing data can reduce the best possible performance of
a predictive model, if the noise of the assumed predictor dependent missingness mechanism is lower than the measurement noise."?
I find this not very surprising. Adding correct information with higher certainty should always be better. Adding correct certain models of the missingness mechanism is of course preferable to high noise imputation.
Why did the authors impute the data for RF? If C4.5 is used than the missingness can be handled without explicit imputation by change of the impurity function. There are also internal impute proposals: Stekhoven & Bühlmann (2012), "MissForest—non-parametric missing value imputation for mixed-type data", Bioinformatics, 28, 1.

**Limitations:**

see above. I think their proposition fails to emphasize a major assumption to their claim.

---

> ### Author Rebuttal · Authors · 2024-08-06
>
> Thank you for your review. We hope we have addressed your primary concern below, and look forward to any further discussion.
>
> >Proof only works since the unobserved noise for k1 and k2 is chosen as it is... If the assumption is switched or equal signs added the proposition will fall apart. It seems the proposition needs to be far more limited than it is right now.
>
> >In 108 should it than not say "Corollary 3.2 states that perfectly imputing missing data can reduce the best possible performance of a predictive model, if the noise of the assumed predictor dependent missingness mechanism is lower than the measurement noise."?...
>
> Proposition 3.1 and Corollary 3.2 are existence claims, which we prove by construction. It is valid and standard to prove such claims by constructing a single viable Data Generating Process (DGP); such a proof does not imply that DGP is the only case where our claim holds. In fact, in situations like the one you've named, where missingness information is higher noise than the rest of the data, there still exist cases where missingness information is needed to achieve the best possible model - we just chose not to focus on them to keep the proof simple. As an example, we have constructed a case where the noise in the missingness indicator is the greatest (k_1<...<k_2<0.5), but the Bayes’ optimal model with missing data is still superior to that with perfect imputation:
>
> Consider a case similar to that used for the proof in proposition 3.1, where we add an additional variable $X_3$ and, as requested, have the noise for the missingness be higher than any other noise.
>
> $$Y = |X_1X_2 - \epsilon_1|, \epsilon_1 \sim \textrm{Bern}(\frac{1}{6})$$
> $$M = |Y - \epsilon_2|, \epsilon_2 \sim \textrm{Bern}(\frac{1}{4})$$
> $$X_3 = |Y - \epsilon_3|, \epsilon_3 \sim \text{Bern}(\frac{1}{5})$$
>
> We also adjust the probabilities for $X_1$ and $X_2$ being true so that this is a balanced classification problem: $X_1, X_2 \sim \textrm{Bern}(\frac{1}{\sqrt{2}})$, so $X_1X_2 \sim \textrm{Bern}(\frac{1}{2})$
>
> The complete proof for this case is available at the bottom of this rebuttal.
>
> >Why did the authors impute the data for RF? If C4.5 is used than the missingness can be handled without explicit imputation by change of the impurity function. There are also internal impute proposals: Stekhoven & Bühlmann (2012), "MissForest—non-parametric missing value imputation for mixed-type data", Bioinformatics, 28, 1.
>
> Avoiding explicit imputation is a valid alternative baseline, which we have added, as shown in Figure 1 of our shared response. The results of this experiment align with those for previous baselines: random forests and decision trees without imputation do not outperform M-GAM.
> Additionally, we note that MissForest is used as an imputation method in Section E.2 of the supplement. We find that MissForest imputation (including MissForest used with a random forest model) does not outperform any of our other impute-then-predict random forest baselines.
>
> Weaknesses:
> >I would have run 2 distinct experiments at least. One with synthetic missingness where the ground truth is known and the missigness can be increased. Additionally I would have liked to see also a mix of MCAR and MAR to demonstrate the results when the assumed mechanism is incorrectly chosen.
>
> We don’t actually make any strict assumptions about the missingness mechanism. We regularize towards MCAR through our $\ell_0$ regularization, but this is not a strict assumption (for example, we still handle the MAR setting in figure 3). As per your suggestion to include MCAR missingness, we’ve included a plot where we switch to MCAR instead of MAR missingness for the experiment done in figure 3 (plot included in the general response). We still handle missingness as well as our competitors, though there is no informative missingness that we can take advantage of in this setting.
>
> >The authors emphasize the interpretability enabled by M-GAM and figures are shown. It would have been nice to discuss the interpretation in the text.
>
> We agree that additional discussion of how to interpret an M-GAM would be appropriate – we will add more discussion around Figure 2, and several more visualizations will be added to the supplement.
>
> ---
>
> Proof:
>
> The bayes optimal model with perfect imputation of $X_1$, and no access to $M$, is still just to predict in accordance with $X_1X_2$ for $P(X_1X_2=Y)=\frac{5}{6}$. When $X_1X_2=X_3$, all information available suggests $X_1X_2$ is correct.
> When $X_1X_2\neq X_3$, we still have the bayes optimal prediction aligning with $X_1X_2$: $P(Y=1|X_1X_2=1,X_3=0) =\frac{5}{9}>0.5$ and $P(Y=0|X_1X_2=0,X_3=1)=\frac{5}{9}>0.5$.
>
> If we instead only have access to $X_1$ when it is not missing, but we also know when $X_1$ is missing (i.e. we know $M$), then the following approach will perform better than the above model:
> $$Y=\begin{cases}
> X_3,&\textrm{if $M=1$}\\\\
> X_1X_2X_3,&\textrm{if $M=0$}
> \end{cases}$$
> When $M=1$, we make additional errors relative to the previous approach at rate
> $$P(M=1)(P((X_3)\neq Y)-P((X_1X_2)\neq Y))=\frac{1}{2}(\frac{1}{5}-\frac{1}{6})=\frac{1}{60}$$
> When $M=0$, we improve our classifier's accuracy by:
> \begin{align*}
> &P(\text{Imputation model is wrong, model with missingness is right}|M=0)\\\\
> &-P(\text{Imputation model is right, model with missingness is wrong}|M=0)\\\\
> =&P(X_1X_2\neq Y=X_1X_2X_3)-P(X_1X_2=Y\neq X_1X_2X_3)\\\\
> =&P(X_1X_2=1,X_3=0,Y=0,M=0)-P(X_1X_2=1,X_3=0,Y=1,M=0)\\\\
> =&P(X_1X_2=1)P( Y=0|X_1X_2=1)P(X_3=0|Y=0)P(M=0|Y=0)\\\\
> &-P(X_1X_2=1)P(Y=1|X_1X_2=1)P(X_3=0|Y=1)P(M=0|Y=1)\\\\
> =&\frac{1}{2}\frac{1}{6}\frac{4}{5}\frac{3}{4}-\frac{1}{2}\frac{5}{6}\frac{1}{5}\frac{1}{4}\\\\
> =&\frac{7}{240}\\\\
> \>&\frac{1}{60}
> \end{align*}
> So the model that uses missingness outperforms the imputation model.

---

> > ### Comment · Reviewer_i65A · 2024-08-13
> >
> > Yes it is formally correct that one can construct situations in which even perfect imputation can be outperformed. As the authors note in cases where the noise of the features is higher than the noise of the missingness model. Also when M is 0 or 1 with a 50/50 chance as seen in the rebuttal. I still find these situations not realistic and having very restricted practical use. Many approaches base on the idea that a missing feature is not essential for the task and information is still covered by the remaining observations accepting some increase in uncertainty. I appreciate the reviewer added such methods in the comparison. Furthermore,  I suggested making a fully synthetic experiment with controlled and increasing missingness in a toy situation where one can easily follow the method and its interpretation. Only as a second part of the sentence the mix of MCAR and MNAR was mentioned, which was less important than the first part. Synthetic missingness in real data is always a mix for which the influence and sensitivity of the method is less accessible. This has not been addressed.
> > However, besides finding the existence claims on the trivial and less interesting side, I see a value in the method itself, also by the discussion with the other reviewers, so I will hire my score accordingly.
> > I would like to point the authors to censoring in Survival analysis as potential interesting application area for which missing data and its interpretation is really important. There the data sets are small, very imbalanced and missigness can be high due to the longitudinal and medical nature.

---

> > > ### Author Response · Authors · 2024-08-13
> > >
> > > We appreciate you taking the time to review our response, and for increasing the score.
> > >
> > > We’re sorry that you find this setting unrealistic, but we are unaware of any standard definition for what a realistic data generating process looks like. We have shown that, under a variety of different constraints, it’s possible to show that the proposition holds. We've shown that it holds in the most realistic setting we can imagine (missingness < 50%, signal from missingness weaker than from other variables), and include the proof at the bottom of this response. We hope this helps show that the setting for this proposition really is quite broad, even though we have not formally quantified this.
> > >
> > > We do believe that, generally, settings with informative missingness are realistic - consider, for example, the result of a medical test being missing because a doctor judged that the patient was not at sufficient risk to need that test; on a survey about politics, an individual skipping questions because they distrust the pollster’s political leaning; or in loan default prediction, a credit report being missing because an individual has no credit history.
> > >
> > > We apologize for missing your request that we add synthetic data - we misunderstood the suggestion, and thought that semi-synthetic data with a ground truth missingness mechanism would satisfy your concern. With this clarification, we’re happy to add fully synthetic examples to our paper (although we aren’t able to include those plots today because of the rules concerning the discussion period).
> > >
> > > Thank you for the suggestion regarding survival analysis – that does sound like a very relevant setting, and we hope to explore it in the future!
> > >
> > > Again, thank you for your feedback and discussion!
> > >
> > > ------
> > >
> > > Proof:
> > >
> > > Consider a case similar to that used for the proof in proposition 3.1, where we add an additional variable $X_3$ and have the noise for the missingness be higher than any other noise.
> > >
> > > $$Y = |X_1X_2 - \epsilon_1|, \epsilon_1 \sim \textrm{Bern}(\frac{1}{12})$$
> > > $$M = \begin{cases} |Y - \epsilon_2|, \epsilon_2 \sim \textrm{Bern}(\frac{1}{4}), \&\textrm{with probability $\frac{1}{2}$} \\\\
> > > 0, \&\textrm{with probability $\frac{1}{2}$}\end{cases}$$
> > > $$X_3 = |Y - \epsilon_3|, \epsilon_3 \sim \text{Bern}(\frac{1}{11})$$
> > >
> > > We also adjust the probabilities for $X_1$ and $X_2$ being true so that this is a balanced classification problem: $X_1, X_2 \sim \textrm{Bern}(\frac{1}{\sqrt{2}})$, so $X_1X_2 \sim \textrm{Bern}(\frac{1}{2})$
> > >
> > > Note that we now also have missingness at well under 50\% of the data (missingness happens a quarter of the time)
> > >
> > > The bayes optimal model with perfect imputation of $X_1$, and no access to $M$, is still just to predict in accordance with $X_1X_2$ for $\mathbb{P}(X_1X_2=Y) = \frac{11}{12}$. When $X_1X_2 = X_3$, all information available suggests $X_1X_2$ is correct.
> > > When $X_1X_2 \neq X_3$, we still have the bayes optimal prediction aligning with $X_1X_2$: $\mathbb{P}(Y=1|X_1X_2=1,X_3=0) = \frac{11}{21} > 0.5$ and $\mathbb{P}(Y=0|X_1X_2=0,X_3=1) = \frac{11}{21} > 0.5$.
> > >
> > > If we instead only have access to $X_1$ when it is not missing, but we also know when $X_1$ is missing (i.e. we know $M$), then the following approach will perform better than the above model:
> > > $$Y = \begin{cases}
> > >     X_3, &\textrm{if $M=1$} \\\\
> > >     X_1X_2X_3, &\textrm{if $M=0$}
> > > \end{cases}$$
> > >
> > >
> > > When $M=1$, we make additional errors relative to the previous approach at rate
> > >
> > > \begin{align*}
> > > \mathbb{P}(M=1)(\mathbb{P}((X_3)\neq Y) - \mathbb{P}((X_1X_2)\neq Y))\\\\
> > >     \&=\frac{1}{4} (\frac{1}{11}- \frac{1}{12})\\\\
> > >     \& = \frac{1}{528}
> > > \end{align*}
> > >
> > > When $M=0$, we improve our classifier's accuracy by:
> > > \\begin{align*}
> > > \&\mathbb{P}(\text{Imputation model is wrong, model with missingness is right and }M=0)\\\\
> > > \&- \mathbb{P}(\text{Imputation model is right, model with missingness is wrong and }M=0)\\\\
> > > = \&\mathbb{P}(X_1X_2 \neq Y = X_1X_2X_3, M=0) - \mathbb{P}(X_1X_2 = Y \neq X_1X_2X_3, M=0)\\\\
> > > = \&\mathbb{P}(X_1X_2=1, X_3=0, Y=0, M=0) - \mathbb{P}(X_1X_2=1, X_3=0, Y=1, M=0)\\\\
> > >     = \&\mathbb{P}(X_1X_2=1)\mathbb{P}( Y=0|X_1X_2=1)\mathbb{P}(X_3=0|Y=0)\mathbb{P}(M=0|Y=0) \\\\ \&- \mathbb{P}(X_1X_2=1)\mathbb{P}( Y=1|X_1X_2=1)\mathbb{P}(X_3=0|Y=1)\mathbb{P}(M=0|Y=1)\\\\
> > >     = \&\frac{1}{2}\frac{1}{12}\frac{10}{11}\frac{7}{8} - \frac{1}{2}\frac{11}{12}\frac{1}{11}\frac{5}{8}\\\\
> > >     = \&\frac{15}{2112}\\\\
> > > \> \&\frac{1}{528}
> > > \\end{align*}
> > >
> > > So the model that uses missingness outperforms the imputation model.

---

### Author Rebuttal · Authors · 2024-08-06

We thank all the reviewers for their thoughtful reviews. We have responded to each reviewer separately. Attached to this message, please find figures used in responses to individual reviewers.

We look forward to further discussion in the days to come.

---

### Decision · Program_Chairs · 2024-09-25

**Decision:**

Accept (poster)

**Comment:**

Paper proposes a novel variant of generalized additive model (GAM) for missing features (M-GAM) to provide comparable accuracy to popular imputation approach, keeping computational cost slow and producing sparse and interpretable model. Both theoretical and experimental analysis is done, showing the usefulness of the proposed model in certain benchmark problems.

Based on comments and scores, all the reviewers are on the positive side of margin. The strengths of the paper are related to novel and useful idea (Reviewers i65A, fLWT, 63xg, W4vf), well-written and structured content (Reviewers i65A, fBin, 63xg, W4vf), valuable method/model (Reviewer i65A), e.g., for practitioners (Reviewers fBin), and extensive experiments (Reviewers fLWT and W4vf). On weaknesses/limitations side, there have been some critics on the experiments (in general Reviewer fBIN, baseline methods Reviewer 63xg,
and e.g., related to missing of synthetic examples with ground truth and where missingness is controlled, Reviewer i65A), performance of proposed method (Reviewers fLWT and fBin), detailed discussion of interpretability of the model (Reviewers i65A and fBin), and theoretical background / claims (Reviewer i65A).

Authors have provided some response to critics in rebuttal, including: 1) additional experiments against baselines of decision trees (DT) and random forest (RF),  and AUC validation metrics for imbalanced datasets (in global rebuttal file, Figure 1), 2) additional computational time analysis of DT and RF (in global rebuttal file, Figure 2), and 3) additional MCAR experiment (in global rebuttal file, Figure 3). Furthermore, more discussion of interpretability around Figure 2 in original manuscript (in discussion with Reviewer i65A) and discussion about DT handling missing values to related work (in discussion with Reviewer 63xg) are to be added. Overall, paper brings valuable information to community and interesting tool for practitioners. Based on these, I would recommended paper to be accepted as
a poster. Authors are encouraged to revise the final paper, including all the improvements and suggestion raised during the rebuttal discussion.